# Ice shelf fracture parameterization in an ice sheet model

Sainan Sun[1], Stephen L. Cornford[2], John C. Moore[3,4], Rupert Gladstone[3], Liyun Zhao[4]

[1] Laboratoire de glaciologie, Université libre de Bruxelles, Brussels, Belgium

[2] Department of Geography, College of Science, Swansea University, Singleton Park, Swansea, SA2 8PP, UK

[3] Arctic Centre, University of Lapland, Rovaniemi, 96101, Finland

[4] College of Global Change and Earth System Science, Beijing Normal University, 100082, China

*Correspondence to*: s.l.cornford@swansea.ac.uk

**Abstract.** Floating ice shelves exert a stabilizing force onto the inland ice sheet. However, this buttressing effect is diminished by the fracture process, which on large scales effectively softens the ice, accelerating its flow, increasing calving, and potentially leading to ice shelf breakup. We add a continuum damage model (CDM) to the BISICLES ice sheet model which is intended to model the localized opening of crevasses under stress, the transport of those crevasses through the ice sheet, and the coupling between crevasse depth and the ice flow field, and carry out idealized numerical experiments examining the broad impact on large scale ice sheet and shelf dynamics. In each case we see a complex pattern of damage evolve over time, with an eventual loss of buttressing approximately equivalent to halving the thickness of the ice shelf. We find that it is possible to achieve a similar ice flow pattern using a simple rule of thumb: introducing an enhancement factor ~10 everywhere in the model domain. However, spatially varying damage (or equivalently, enhancement factor) fields set at the start of prognostic calculations to match velocity observations, as is widely done in ice sheet simulations, ought to evolve in time, or grounding line retreat can be slowed by an order of magnitude.

## 1 Introduction

The largest uncertainties in sea level rise prediction are the dynamic ice sheet contributions (Jevrejeva et al., 2016). Two recent ice sheet model studies came to surprisingly different conclusions regarding the Antarctic contributions to 21st and 22nd century sea level rise: Ritz et al. (2015) concluded that Antarctic contributions of a meter or more are implausible, whereas DeConto and Pollard (2016) find that not only is this plausible, but should even be considered likely if the previously underappreciated hydrofracturing process is included and linked to climate dynamics. Observations show an increased rate of ice discharge from Greenland and Antarctica in recent decades (Shepherd et al., 2012). Calving is directly

responsible for a mass loss comparable to that from ice shelf basal melting (Rignot et al., 2010; Depoorter et al., 2013; Liu et al., 2014). Furthermore, increased rates of calving and basal melt seem intertwined and act in concert to enhance mass loss from ice shelves that are in negative mass balance under the present climate (Liu et al., 2014; Åström et al., 2014). Mass loss from ice shelves does not contribute to sea level rise directly, but via the restraint ice shelves apply to the ice discharge from

inland to ocean across the grounding line; in other words mass loss from ice shelves is expected to weaken their buttressing effect. Fürst et al. (2016) show that ice shelves in the Amundsen and Bellingshausen seas are thought to be more vulnerable to calving events than other regions meaning that even a small amount of increased calving can trigger dynamical responses that raise sea level. Thus to better predict the future evolution of the ice sheets, processes related to calving should be better understood and described.

Macro-scale fractures originate from micro-scale cracks, which are more likely to form when stresses within the ice exceed a few hundred kPa. On microscopic scales fracturing is a discrete process which operates on time scales determined by the speed of sound in ice, and the rupture speed could be influenced by the local variations of stress state and material properties (Ye et al., 2016). Combining these processes in a single model presents very difficult numerical challenges that have only been attempted for a few cases using discrete element models (Bassis and Jacobs, 2013; Åström et al., 2013; Åström et al.,

2014). This discrete approach has a firm basis in physics with, for example, Glen's flow law emerging naturally. On macroscopic scales, the development of cracks may be seen as a continuum and long-term process. The net effect is to soften ice, and hence potentially accelerate ice flow. Fractures form as the damage effect accumulates. Propagation and penetration of fractures trigger calving events. Previous continuum studies of calving fall into two main categories, but all are essentially empirical rather than being based on fundamental fracture process physics. Models of the first type treat

calving as the result of macro-scale crevasses, but do not consider any direct coupling between crevasses and the flow field. Crevasse depth depends on the stress field (Nye, 1957; Benn et al., 2007; Weertman, 1973), but the stress field is affected in turn only by the change in forcing that immediately follows a calving event. Models in this category (Nick et al., 2010; Nick et al., 2013; Cook et al., 2014), compute crevasse depth based on instantaneous fields, and hence do not take into account the stress history in the development of fractures. Models in the second category use a Continuum Damage Mechanics (CDM)

approach, which treats calving as a continuum process that develops from micro-scale cracks to macro-scale crevasses, and damage has an effect on the viscous behaviour of ice flow (Pralong and Funk, 2005; Jouvet et al, 2011; Duddu and Waisman, 2012; Borstad et al., 2012; Borstad et al., 2013; Albrecht and Levermann, 2014; Krug et al., 2014; Bassis and Ma, 2015). Here we propose a continuum damage model (CDM) which considers conservation of damage during ice flow as well as local sources of damage. The local source of damage comes from the stress field and water depth in crevasses following the

physical mechanism of Nick et al (2010), while a conservation equation describes the evolution of the vertically integrated damage field due to advection, stretching and mass loss or gain from the glacier's upper and lower surfaces. The

development of damage has an impact on ice viscosity, and therefore influences the evolution of ice flow through Glen's flow law, so that the damage and flow fields are strongly coupled. Potentially important phenomena such as the detailed accretion of marine ice within basal crevasses, and the necking phenomena of Bassis and Ma (2015) are not included: our aim here is to construct a model that is amenable to large scale calculations and investigate the broad outline of its impact on ice flow.

We use the adaptive mesh refinement model BISICLES (Cornford et al., 2013) as our ice flow model, and add the CDM to it. The performance of the CDM is tested on the MISMIP+ (Marine Ice Sheet Model Intercomparison Project) experiments (Asay-Davis et al., 2016), which are based on an idealized marine ice sheet that has strong lateral stresses (Gudmundsson et al., 2012). A range of steady-state and time dependent simulations were carried out with and without the CDM in an effort to answer four related questions. How does damage influence the evolution of the ice sheet, in particular the behaviour of the grounding line, and the ice velocity field? Can we achieve the same steady state location of the grounding line in the model with or without the CDM component by making a rule of thumb adjustment to the (uniform in this case) rheology parameter $A$, which is typically unknown and so must be tuned to observations? If such an adjustment can result in similar steady states, how does the transient response between the models then differ, when they are subjected to an external forcing that leads to thinning of the ice shelf? Is it necessary to evolve the damage model in time, or is it possible to simply construct a damage field at the start of a calculation and hold it constant throughout the simulation?

In the next Section, we describe the physics used in this study, including the CDM and ice flow model. Then we present in Section 3 a suite of experiments based on MISMIP+ to evaluate the model performance and the effect of damage on ice sheet evolution. We present the results in Section 4 and a discussion in Section 5.

## 2 Model Description

### 2.1 Ice flow governing equations

We implemented the damage model in the marine ice sheet model BISICLES (Cornford et al., 2013). Ice flow velocity is computed by solving a vertically-integrated stress balance equation, in this case the Shallow-Shelf Approximation (SSA) of the Stokes equations (MacAyeal et al., 1996). This ice dynamics formulation performs well for ice shelves and fast-flowing ice stream simulations. Floating ice is assumed to be in hydrostatic equilibrium, so given a bed elevation $b$ and ice thickness $h$, the surface elevation $s$ is

$$s = max\left(h + b, \left(1 - \frac{\rho_i}{\rho_w}\right)h\right), \quad (1)$$

where $\rho_i = 910\ kg\ m^{-3}$ and $\rho_w = 1028\ kg\ m^{-3}$ are densities of ice and ocean water.

Ice thickness $h$ and horizontal velocity $\boldsymbol{u}$ satisfy the mass conservation equation

$$\frac{\partial h}{\partial t} + \nabla \cdot [\boldsymbol{u}h] = a - M \qquad (2)$$

and the two-dimensional stress-balance equation

$$\nabla \cdot [h\bar{\mu}(2\dot{\boldsymbol{\epsilon}} + 2tr(\dot{\boldsymbol{\epsilon}})I)] + \tau^b = \rho_i g h \nabla s \qquad (3)$$

together with lateral boundary conditions. In equation (2), $a$ is the surface ice mass accumulation and $M$ is the basal melt rate

of the ice shelf. In equation (3), $tr$ is the trace operator, $\dot{\boldsymbol{\epsilon}}$ is the horizontal strain rate tensor,

$$\dot{\boldsymbol{\epsilon}} = \frac{1}{2}[\nabla \boldsymbol{u} + (\nabla \boldsymbol{u})^T], \qquad (4)$$

$I$ is the identity tensor and $\tau^b$ is the basal friction. Basal friction is computed according to Tsai et al. (2015), tending to a

power law far upstream from the grounding line, and to a Coulomb friction law near the grounding line. $h\bar{\mu}$ is the vertically

integrated effective viscosity, which would conventionally be given by Glen's law, but is modified here to include an

additional damage parameter (see section 2.2).

BISICLES is constructed using the Chombo parallel AMR (Adaptive Mesh Refinement) framework, which allows us to use

a non-uniform, evolving mesh during simulations. Here, we implement three levels of local refinement on top of a coarse

mesh (level zero) spanning the domain at 4 km resolution. The grid cell size is refined by a factor of 2 at each refinement

level leading to a finest resolution of 0.5 km around the grounding line. The time step size satisfies the Courant-Friedrichs-

Lewy condition everywhere, meaning, for the geometry here, about 16 time steps per year.

**2.2 Damage model**

We construct a vertically integrated damage model by treating the ice sheet as having upper and lower layers of ice entirely

fractured by surface and basal crevasses respectively, and an undamaged central layer (Fig, 1). Therefore, the scalar damage

variable, $D(x, y, z) \in [0,1)$ employed in vertically varying models (Pralong and Funk, 2005; Jouvet et al, 2011; Keller and

Hutter, 2014; Krug et al 2015; Bassis and Ma, 2015; Mobasha et al, 2016) takes on either the value 0 (in the central layer) or

1 (in the upper and lower layers). The principal damage variable in our model is $d(x, y) \in [0, h(x, y))$, the vertical integral

of $D(x, y, z)$, and our closest analogue to the usual $D$ is its vertical average, $\bar{D}(x, y) = d(x, y)/h \in [0,1)$.

Damage enters the stress balance equation through a modification to Glen's law. The usual relationship between deviatoric

stress $\boldsymbol{\tau}$ and rate-of-strain $\dot{\boldsymbol{\epsilon}}$ is:

$$2 A \tau^2 \boldsymbol{\tau} = \dot{\boldsymbol{\epsilon}}, \qquad (5)$$

where the flow rate exponent $n = 3$, and $A$ is a flow rate factor, which is typically temperature dependent but set to be constant here. We replace this with

$$2 A \tau^2 \boldsymbol{\tau} = (1 - D(\tau))^3 \dot{\boldsymbol{\epsilon}}, \quad (6)$$

which – given the shallow shelf approximation – results in an expression for the vertically integrated effective viscosity,

$$2 h \bar{\mu} = [h - d(\tau_1)] A^{-\frac{1}{3}} \dot{\epsilon}^{-\frac{2}{3}}. \quad (7)$$

A relationship between damage and the first principal stress, $d(\tau_1)$, must be identified, at which point the stress balance equation can be solved numerically for $d$ and $\boldsymbol{u}$ together. Specifying the damage-stress relationship in this way assumes that damage evolves on a similar or faster timescale to the ice velocity field. Many authors specify instead a damage evolution rate, which, in e.g. Krug et al (2014), and given typical stresses in an ice shelf, amounts to a timescale of around 1 year.

Notice that damage affects only the deviatoric stress (as in Jouvet et al, 2011 and Krug et al, 2014) and does not affect the gravitational driving stress. We might expect such a modification if we had instead modified the full Cauchy stress (as in Pralong and Funk, 2005, Bassis and Ma, 2015, and Mobasha et al 2016), but have assumed that damage has no impact with respect to isotropic compression or vertical shear, so that the usual hydrostatic vertical stress balance, and the usual vertical integral of the resulting horizontal pressure gradient holds. This is analogous to assuming that the crevasses are filled with an inviscid material having the same density as ice.

In the absence of advection, we could now prescribe $d(\tau_1)$ by equating it to the total depth of crevasses, computed following Nye (1957), Benn et al (2007), and Nick et al (2010). There, the depth of a mode I crevasse at the upper surface is:

$$d_s = \frac{\tau_1}{\rho_i g} h + \frac{\rho_w}{\rho_i} d_w, \quad (8)$$

while at the lower surface it is

$$d_b = \frac{\rho_i}{\rho_w - \rho_i} \left( \frac{\tau_1}{\rho_i g} h - h_{ab} \right), \quad (9)$$

where $d_w$ is the water depth in the surface crevasse and $h_{ab}$ is the thickness above floatation. The total local crevasse depth is then $d_l = \max(d_s, d_s + d_b, 0)$.

A water depth $d_w \sim h/2$ is in fact required in Nick et al (2010) for any calving to take place at all, and would clearly have a substantial impact on our calculations too, but for this paper we consider only dry crevasses, with $d_w = 0$. We also ignore any lower limit on the stress needed to open a crevasse, so that we will tend to produce small crevasses where there should be none. As we will see in the results, the major impact of the damage model is in the ice shelf and around the grounding line, where large tensile stresses readily exceed such limits.

Inasmuch as a shallow shelf model is a good approximation to the full Stokes model, our choice of the Nye zero stress model above is similar to the long-term behaviour of Krug et al (2014), at least for surface crevasses. In that model, damage grows to $D \to 1$ as $t \to \infty$ where and only where the Cauchy stress is tensile, just as in the Nye model, giving the depth of surface

crevasses. Basal crevassing could be included in such a model by adding the water pressure to the Cauchy stress, as in Keller and Hutter (2014).

One more modification is needed to reflect the transport of damage by ice flow. At any one time and place $(x, y, t)$ we would have two fields, the $d_l(x, y, t)$ computed above, and a field of transported crevasse depths $d_{tr}(x, y, t)$ which would have originated at $(x', y', t' < t)$ and been carried downstream, stretched, compressed, and so on. We assume that crevasse closure does not result in bonding of the crevasse surfaces, at least on the timescale of the closure itself, so that the final relationship $d(\tau_1)$ is given by

$$d(\tau_1) = \max(\, d_l(\tau_1), d_{tr}\,). \quad (10)$$

This means that regions of the ice shelf under lower stress inherit damage from any higher stress region upstream. The transported total crevasse depth $d_{tr}$ is found by solving a damage transport equation

$$\frac{\partial d_{tr}}{\partial t} + \nabla \cdot (\boldsymbol{u} d_{tr}) = -[\max(a, 0) + \max(M, 0)] \frac{d_{tr}}{h}. \quad (11)$$

The left hand side of Eqn. (11) expresses the conservation of the vertically integrated damage under ice flow, and includes both the motion of crevasses with the flow ($\boldsymbol{u} \cdot \nabla \mathrm{d}_{tr}$), and stretching and compression ($d_{tr} \nabla \cdot \mathbf{u}$), which, all else being equal, holds the ratio $d_{tr}/h$ constant in a diverging or converging horizontal flow field. The right hand side assumes that accumulation at the upper surface ($a$) increases the thickness of undamaged ice, while basal melting ($M$) erodes the crevassed underside, so that both terms cause a reduction in vertically integrated damage for the cases considered here, where $a > 0$ and $M > 0$. Note that we do not attempt to include any additional accretion or ablation physics particular to the inside of crevasses. We specify Dirichlet conditions $d_{tr} = 0$ on all inflow boundaries, while initial conditions are determined by evolving the coupled system to steady states, starting from the initial guess $d_{tr}(t = 0) = d_l(t = 0)$. We also set $d_{tr}(t - \Delta t) = d(t - \Delta t)$ at every time step, which in effect imposes a new initial condition to ensure that $d_{tr}(t - \Delta t) \geq d_l(t - \Delta t)$ everywhere. Note also that Eqn. (11) has no explicit healing term to represent the effect of overburden pressure, which will lead us to overstate the damage field. We assume it is relatively unimportant compared to crevasse opening in the largely tensile ice shelf flow fields most affected by damage in the results presented here.

## 3 Experimental design

To answer the questions mentioned in Section 1, we carried out a set of numerical experiments based on the third Marine Ice Sheet Model Intercomparison Project (MISMIP+, Asay-Davis et al., 2016). MISMIP+ includes a number of experiments based on an idealized marine ice sheet geometry derived from Gudmundsson (2012) and Gudmundsson (2013). Ice flows along an 800 km long and 80 km wide submarine bedrock trough, from an ice divide at one end to an ice shelf and calving front at the other. The geometry features an ice shelf with lateral stresses that buttress upstream ice to the extent that it is

possible to obtain a stable equilibrium with its grounding lines positioned on a retrograde bedrock slope, that is, a slope rising in the direction of ice flow. The suite of simulations based on different models and basal melt rates are summarized in Table 1 and described in detail below.

We investigated equilibrium states by carrying out the MISMIP+ no ocean forcing experiment (Ice0) with and without the damage model. Ice0 requires that models are close to steady state with a grounding line crossing the centre of the trough (y = 40 km) at around x = 450 km, and that they demonstrate this by showing insignificant grounding line migration over 100 years in the absence of ice shelf basal melt. Without the damage model, we set $A = 2.0 \times 10^{-17} \, Pa^{-3} \, a^{-1}$, just as in Asay-Davis et al (2016), and found a steady state. We then switched on the damage model and ran for 1000 years to assess grounding line migration due to the weakened ice shelf. We will refer to this simulation as IceD (in general $D$ indicates inclusion of the damage model). In order to start the MISMIP+ experiments from the required grounding line location at x = 450 km, we run a series of IceD simulations with different values of the rate factor $A$. For each value of $A$, a new steady state grounding line location is obtained, and we select the value $A'$ for which the location is closest to the originally required grounding line at x = 450 km. We will refer to this steady state experiment as IceD0.

Once IceD0 had been completed, we carried out the remaining MISMIP+ experiments with the given values of $A$ and $A'$. Ice1r and IceD1r see the models respond to a simple basal melt formula that concentrates ablation close to (but not at) the grounding line as it evolves over 100 years,

$$M = \frac{1}{5} tanh\left(\frac{H_c}{75.0}\right) max\big((100 - Z_d), 0\big) \quad (12)$$

where $H_c$ is the water column thickness and $Z_d$ is ice shelf draft. Ice1ra and IceD1ra see the ice sheet change over the next 900 years when the basal melt rate is set to zero, while Ice1rr and IceD1rr continue for the next 900 years under the influence of basal melt (Eqn. 12). Ice2r, IceD2r, IceD2ra, IceD2ra, Ice2rr and Ice2Drr follow the same general pattern, but impose a different melt rate

$$M = \begin{cases} 100 \, m \, a^{-1}, & x \geq 480 \, km \\ 0, & x < 480 \, km \end{cases} \quad (13)$$

where x represents the distance away from the ice divide. This melt rate is concentrated away from the grounding line and does not evolve with it, allowing a thick ice shelf to form in the wake of a retreating grounding line. This high basal melt rate at the ice front is designed to simulate the effect of mass loss far from the grounding line on ice flux, which is an analogue to calving events.

Finally, we carried out versions of the Ice0, Ice1r, and Ice1ra experiments with the same $A'$ and initial damage field as in IceD0, but without allowing the damage field to evolve over time. We will refer to these experiments as IceF0 and IceF1r, with the 'F' standing for fixed-in-time damage. This resembles realistic cases (e.g. Gong et al., 2014; Favier et al., 2015;

Cornford et al., 2015; Sun et al., 2014) where an initial damage field is estimated to match observations of velocity in the ice shelf at the start of a simulation, and held constant thereafter.

## 4 Results

### 4.1 Experiments IceD and IceD0

Switching the damage model on given the steady geometry of Ice0 with $A$ produces widespread weakening of the ice shelf, resulting in 100 km of grounding line retreat. Fig. 2 shows how the damage field evolves for 1000 years from the moment the damage mechanism begins. On grounded ice, the vertically averaged damage, $\bar{D} = d/h$ is generally low, which can be attributed to both the low viscous stress and the ice overburden acting against basal crevasse formation. In the ice shelf, $\bar{D}$ is typically about 1/2, varying from around 1/3 close to the grounding line in the center of the channel, to nearly 1 where the

grounding line crosses the channel walls. As time passes, damage is advected downstream so that strips of ice with $\bar{D}$ close to 1 extend all the way from the grounding line to the calving front. Meanwhile, the acceleration caused by this ice shelf weakening results in the grounded ice thinning and in turn the grounding line retreats, by around 70 km in the first 100 years of the simulation and a further 30 km over the full 1000-year simulation.

Increasing the stiffness of ice by reducing $A$ to $A' = 1.5 \times 10^{-18} \, Pa^{-3} \, a^{-1}$ is sufficient to counter the damage and hold

the grounding line steady at x = 450 km. Such an order of magnitude change in $A$ corresponds to an approximately 20 K difference in ice temperature, or, more pertinently, the introduction of an enhancement factor ~ 10 in the model without damage. Fig. 3 shows the vertically averaged damage $\bar{D}$ at the end of the IceD0 experiment, computed with the damage model and the lower rate factor $A'$. At this point the model is close to steady state, having been allowed to evolve for 30,000 years: neither the grounded area nor the volume above flotation changes substantially over the course of 1000 years.

Just as before, ice far upstream from the grounding line is not strongly damaged, whereas $\bar{D}$ is typically around 1/2 in the ice shelf, and larger close to the domain boundaries. It is also apparent from Fig. 3 that $\bar{D}$ begins to grow over a region a few kilometers upstream from the grounding line.

Although the pattern of damage and its impact on effective viscosity varies both laterally and between grounded and floating ice, the net effect is to produce a grounding line that is rather similar in shape and position to that of Ice0 (Fig. 4). Having

established that it is possible to 'emulate' the damage model, at least in steady state, by a simple (uniform) change to $A$, it is natural to ask if the same is true when the ice shelf is perturbed.

## 4.2 Experiments IceD1* and IceD2*

Fig. 5 shows the evolution of $\bar{D}$ and the grounding line location over time in the IceD1r and IceD1ra simulations. IceD1r sees the grounding line in the centre of the channel retreat from around x = 450 km to x = 390 km over the course of 100 years while the basal melt rate (Eqn. 12) is applied. At the same time, the damage field evolves to maintain a pattern of low damage ($\bar{D} \ll 1$) on the grounded ice and more damage ($\bar{D} \sim 1/2$) in the ice shelf. Much of the ice shelf is thin ($h < 200$ m) so that a high value of $\bar{D}$ implies only a small reduction in buttressing caused by the ice shelf. IceD1ra, which continues from IceD1r, specifies that basal melt rates return to zero, allowing the grounding line to advance toward the IceD0 steady state position by t = 1000. Ice in newly grounded regions has almost no damage, since the advection of damage is significantly faster than grounding line advance. Between t = 100 and t = 1000 years, the formerly heavily damaged region is completely lost downstream of the grounding line and a tongue of less damaged ice ($\bar{D} \sim 1/3$) extrudes from the grounding line and a pattern akin to IceD0 (Fig. 3) is reached by t = 1000.

Fig. 6 plots the damage field and the grounding line during the evolution of ice in IceD2r and IceD2ra experiments. The ice shelf is removed entirely beyond x = 480 km during the first 100 model years. The damage field in the remainder of the ice shelf appears much as it did in the initial state, albeit with a narrow strip of high ($\bar{D} \sim 1$) damage right at the calving front. The grounding line retreats by around 20 km in the center of the channel, while the damage field evolves so that $\bar{D} \sim 1/3$ immediately downstream, growing to $\bar{D} \sim 1/2$ before abruptly increasing at the calving front. Maintaining the same forcing sees the same trend continue, with the grounding line retreating by a further 40 km in 1000 years (not shown here). Experiment IceD2ra, on the other hand, permits the calving front to advance to x = 640 km. It exhibits a travelling front of strong damage, separating thicker ice ($h > 200$ m) which has been carried downstream from the ice shelf of IceD2r from ice which had been permitted to accumulate on the open sea - had such accumulation not been permitted, this damage front would have been coincident with the calving front. Over time, the original pattern of IceD0 is recovered, with $\bar{D} \sim 1/3$ near the grounding line, $\bar{D} \sim 1/2$ in the lateral centre of the shelf for most of its length, and strips where $\bar{D} \sim 1$ close to the domain boundaries.

We compare the grounded area change and ice volume change between different models and ocean forcing in Figs. 7 and 8. After tuning the parameters, our model with the CDM produces similar retreating/advancing trends to the published model results (Asay-Davis et al., 2016). For both BISICLES and BISICLES_D (BISICLES with CDM), some grounded regions become afloat when we implement melt rates, and the grounded area decreases gradually until the melt rates cease. The floating area could be re-grounded if melt rates were no longer applied even on the retrograde bedrock slope. Ice volume change has the same trend as grounded area. The ice volume above floatation (i.e. that can affect sea level) decreases when forced by basal melt or calving, and increases to nearly the initial state when ocean forcing disappears. In both versions of

BISICLES, the ice sheet is more sensitive to basal melt near the grounding line than to extreme high basal melt representing calving at the front. However, the BISICLES_D produces less retreat under both ocean forcing scenarios.

## 4.3 Experiment IceF0

The final experiment contrasts retreat rates between three models: the original model, the model with evolving damage, and a model where the damage is initially the same as in the evolving damage model, but remains constant in time. Fig. 9 shows that all three models maintain a near steady state when not perturbed by ice shelf melting (Ice0, IceD0, IceF0), and indeed, the original and damage-evolving models (Ice1r, IceD1r) show a similar rate of grounding line retreat under ice shelf ablation. However, with the damage field fixed in time, experiment IceF1 shows a far slower rate of retreat – 500 km$^2$ over the course of a century, rather than the 3000 km$^2$ seen in the other two cases. To reiterate: 'emulating' the damage model by simply softening the ice uniformly gives far closer results to the full model than having an initially identical damage field but neglecting to evolve it over time.

## 5 Discussion

The CDM produces a plausible damage distribution on this particular MISMIP+ geometry: damage is observed to be low on the grounded ice, while it increases dramatically when the ice crosses the grounding line and increases gradually downstream from there similar to the crevasses distribution of Pine Island Glacier founded by Bindschadler et al., 2011. The experiments IceD, which in comparison to Ice0, explicitly shows the result of adding damage to the ice shelf, produced rapid grounding line retreat over the full 1000 year simulation (Fig. 2). Thus, the buttressing force of the undamaged ice and its corresponding ice viscosity in the shelf is essential to the stability of the ice sheet when using the MISMIP+ geometry.

Although the damage distribution is highly localized, we showed that by tuning the viscous parameter $A$ in the control experiment IceD0 we can match the grounding line in steady state achieved with Ice0. The required flow law parameter is an order of magnitude greater than the MISMIP+ value of $A$. Simulations IceD1r, IceD1rr, IceD1ra, IceD2r, IceD2ra, and IceD2rr all exhibit similar overall trends to the unmodified BISICLES model, despite extensive spatial variation in the damage field. However, the effective viscosity around the grounding line under the various experiments with and without the damage model is similar. This implies that for this geometry, the viscosity of ice around the grounding line essentially controls the grounding line position, while the state of the ice shelf far from grounding line has much less impact. The ice shelf is relatively thin (<200 m) except near the grounding line and thus contributes little buttressing to inland ice flux.

Including a CDM in a realistic simulation may, perhaps counterintuitively, result in lower sensitivity of the ice sheet to ice shelf ablation and sea level rise. Recall that, given an initial stable state with similar geometry and flow field, BISICLES_D saw lower rates of both retreat and advance in the Ice1 and Ice2 experiments than the BISICLES - especially the Ice2

experiment (Fig. 7, Fig. 8). We can attribute this to the lower value of $A$ (more viscous ice) needed across the domain to compensate for the weaker ice shelf. Once that weaker ice shelf is removed, the remaining ice sheet tends to have a larger viscosity, certainly upstream from the grounding line, than it did in the undamaged case. Realistic simulations, at least if tuned to match observed geometry and velocity, might be expected to behave in the same way.

5 The relative effects on the grounding line position of experiments with 'realistic' basal melt patterns, where basal melt is highest close to (but not at) the grounding line (Ice1r), and those designed to mimic calving (Ice2r), where basal melt is very high near the ice shelf edge, show that calving has much less control on grounding line retreat. This does not mean that calving is in general unimportant for the grounding line position, because in our special geometry, the side walls apply strong back stress and the removed ice is mostly downstream of the side wall, which will not always be the case in reality, in 10 particular for large ice shelves.

The evolution of the damage field in all of the experiments can be approximated, crudely at least, by a simple rule. There is little or no damage to grounded ice, while $\bar{D} \sim 1/2$ at all times in the ice shelf, with lower values ($\bar{D} \approx 1/3$) close to the grounding line and in confined regions of the shelf, unless the ice is so thin as to contribute little buttressing. Thus, in the absence of a damage model, it might be possible to emulate its effects simply by setting $\bar{D} \sim 1/2$ (or equivalently, 15 multiplying $A$ by $\sim 1/8$, that is setting an enhancement factor $E \sim 8$) (Paterson, 2000). The primary cause of this simple pattern is the Nye crevasse formulae (Eqns.8 and 9): basal crevasses are deep – an order of magnitude deeper than surface crevasses – when and only when ice is close to or at flotation, while in a simple model of the ice shelf, with no lateral variation and no buttressing, it is straightforward to see that $d_b + d_s = \frac{h}{2}$ by noting that the vertically integrated stress given by the calving front boundary condition is maintained throughout the shelf. As $d_b \gg d_s$ by about a factor of 10, depending 20 on densities, modelled damage is much higher in the shelf than on grounded ice. This is consistent with observations that large-scale crevasse-like surface features are common on the ice shelves along the Amundsen and Bellingshausen Seas and on the smaller ice shelves between the Amery and Ross ice shelves of east Antarctica (Liu et al., 2015). Ground-penetrating radar show that many of these are, in fact, the surface expression of deep and wide transverse basal crevasses (Bassis and Jacobs, 2013; McGrath, et al., 2012) or longitudinal sub-glacial melt channels (Vaughan et al., 2012), that may penetrate 200 25 m into the base of the ice shelf, while the surface depressions are typically 30 m lower than the usual ice shelf surface (Liu et al., 2014).

While the damage model might be approximated by the ansatz described above, it appears that the practise of estimating a spatially varying damage (or enhancement factor, or rate factor) field by solving an inverse problem to match observed velocities may be problematic, unless the damage evolves in time. An experiment along these lines (IceF1) was the only case 30 where we saw substantially – nearly an order of magnitude – slower retreat rates in the MISMIP+ Ice1 experiment. In fact, the retreat rate was even slower than the typical Ice2r retreat rates, where melt is restricted to a downstream location and a

thick ice shelf persists to buttresses the upstream ice. Since there is little ice shelf left in the IceF1 experiment, it is clear that the damage field close to the grounding line has a major role in the ice shelf dynamics, and neglecting to update it as the grounding line retreats leads to a lower flux $q$ across the grounding line – a result that would be expected by considering the role of the rate factor in approximations to $q$ (Schoof 2007, Tsai 2015).

A damage model with sufficient skill to represent all relevant processes would require further development. An obvious limitation is the choice of a vertically integrated model, when a vertically varying flow field is required to describe processes such a necking (Bassis and Ma 2015), or, even if a vertically integrated flow field is sufficient, a vertically integrated damage model may not be (Keller and Hutter, 2014). It is also clear that the Nye zero stress model cannot be the whole story: if nothing else it requires a phenomenological parameter (the crevasse water depth) to treat calving events, which may in effect be standing in for entirely different physics, such as brittle failure (Krug et al., 2015). It may be important to consider a threshold stress for damage initiations, and mechanisms of crevasse healing (Albrecht and Levermann, 2014).

## 6 Conclusions

We added a continuum damage mechanics model component to the BISICLES ice sheet model. The model computes the evolution of a vertically integrated damage field by generating local damage, and transporting it downstream with the ice flow field. Although the modification to Glen's flow law is based upon the crevasse opening formulae of Nye (1957), Benn et al. (2007) and Nick et al. (2010) it can be adapted to any relationship between local stretching stress and damage.

The model was tested by carrying out the MISMIP+ (Asay-Davis et al 2016) experiments with and without the damage model. Simply introducing the damage calculation results in a much weaker ice shelf given the same flow law parameter ($A$), so that the grounding line retreats. However, realistic simulations tend to tune $A$ to match observations, for example by solving an inverse problem to match velocities. We could produce similar steady states, defined primarily by grounding line position, for the two models by choosing a value of $A$ around ten times lower for the flow model with damage than for the model without. Once we had done so, the response of the ice stream to ablation of the ice shelf was similar in both cases, with the damage model resulting in slightly lower rates of retreat, especially when the ablation was limited to the downstream portion of the ice stream. We explain this lower rate of retreat by noting that damage is far greater in the ice shelf, which must be compensated for by stiffer ice upstream to have the same steady state: once the ice shelf is removed, we are left with the dynamics of stiffer ice, albeit mildly stiffer ice because of the evolution of damage around the grounding line. Put another way, if ice shelves are generally weaker, their loss is of lower consequence.

Although tuning $A$ in a simplistic way may be a plausible alternative to damage modelling, initializing an ice shelf rheology to match observed velocities and then holding that rheology constant in time is not. In this case, although the shelf still

provides less buttressing, once the grounding line begins to retreat we are left with the dynamics of not just mildly stiffer, but far stiffer, ice, so that retreat rates might be underestimated by an order of magnitude.

## Acknowledgements

We thank Lionel Favier for the helpful comments on the former version of this manuscript. This study is supported by China Postdoctoral Science Foundation NO. 212400240, National Key Science Program for Global Change Research (2015CB953601), and National Natural Science Foundation of China (Nos. 41530748, 41506212). Stephen L. Cornford was funded by the UK NERC Centre for Polar Observation and Modelling. Rupert Gladstone is funded by the Academy of Finland under grant number 286587.

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

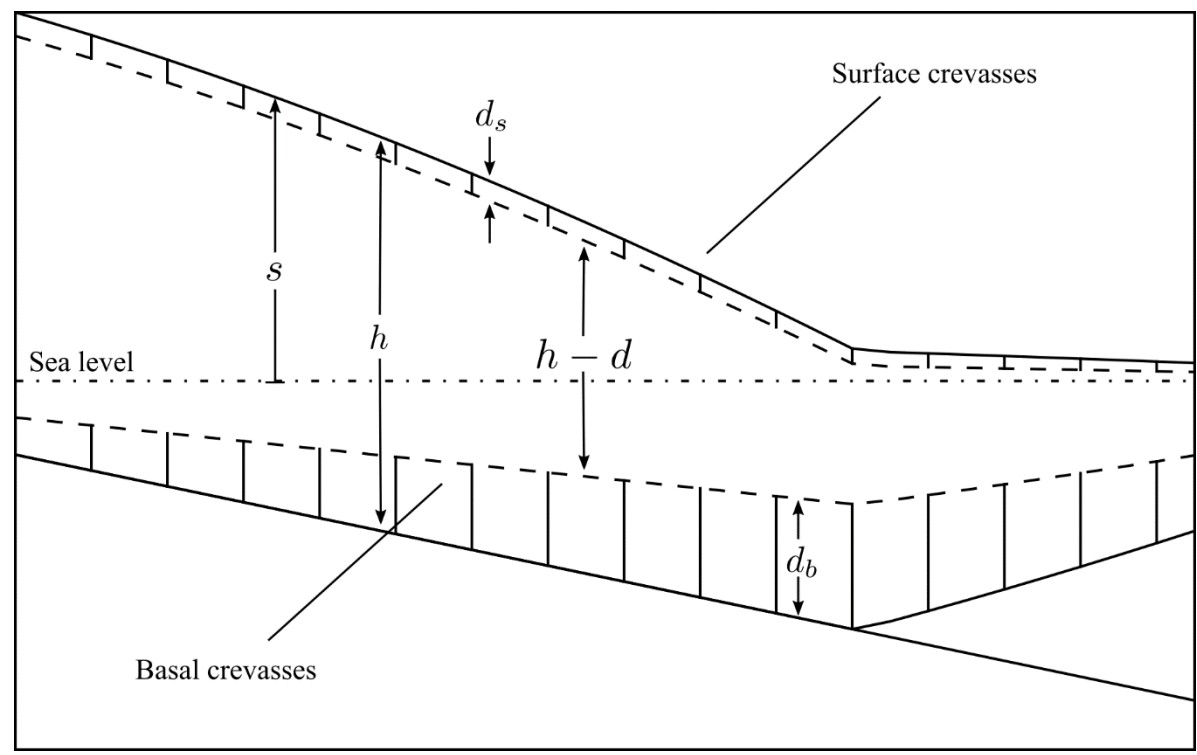

**Figure 1: Sketch of the vertically integrated continuum damage model. The full thickness of the ice sheet ($h$) is divided into three layers. The upper and lower layers are assumed to be entirely riven by surface crevasses of depth $d_s$ and basal crevasses of depth $d_b$, while the layer between them (with thickness $h - d = h - (d_s + d_b)$ is considered intact. The crevassed layers offer no resistance to horizontal longitudinal and lateral shearing, but behave in the same way as undamaged ice with regard to isotropic compression and vertical shear.**

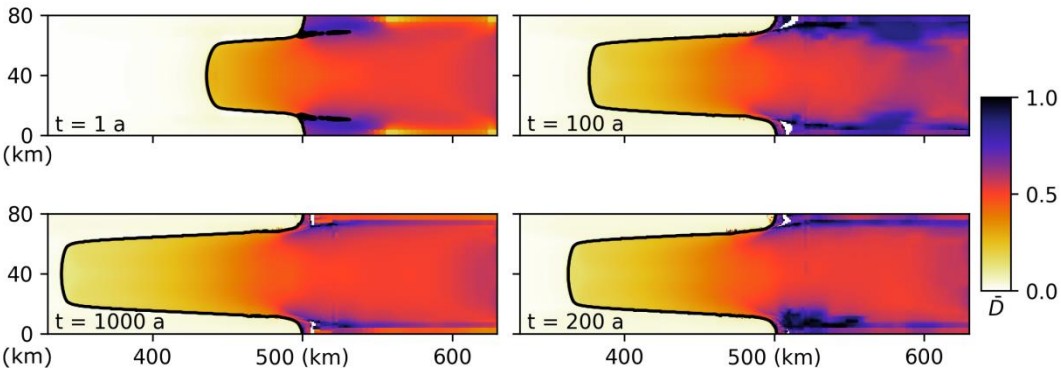

**Figure 2: Evolution of the damage field $\bar{D}$ and the grounding line in IceD, starting from a steady state with $\bar{D} = 0$. The immediate result of 'switching on' the damage model is the creation of a heavily damaged ice shelf (with $\bar{D}\sim 0.5$). Over time, this weaker ice shelf results in the grounding line (solid black contour) retreating over more than 100 km. At the same time spots of intense damage ($\bar{D}\sim 0.9$) generated where the grounding line crosses the channel walls around x = 500 km and y = {10, 70} km are transported downstream, to form strips that extend to the calving front.**

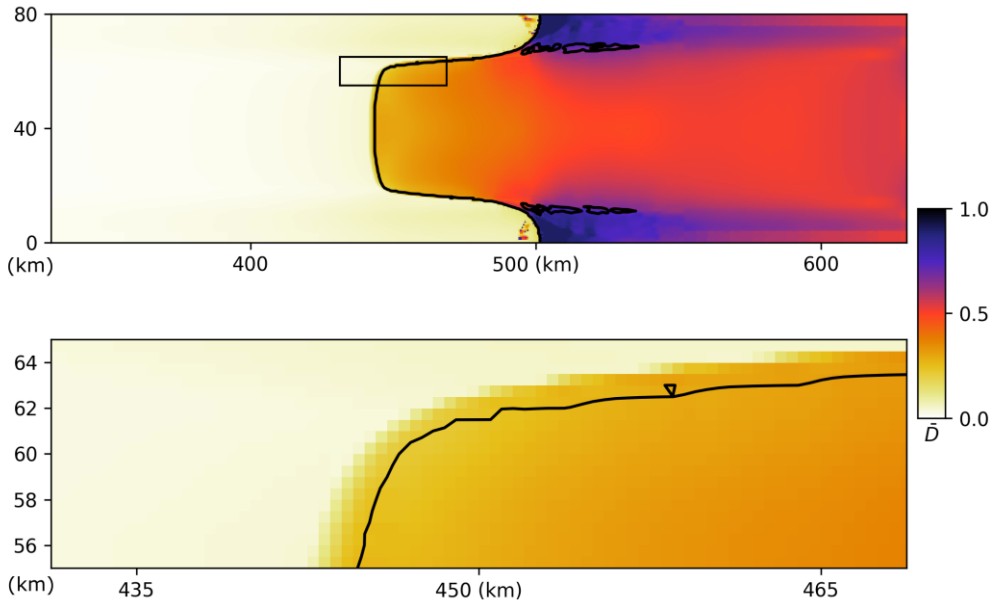

**Figure 3: Vertically averaged damage field $\bar{\bar{D}}$ in experiment IceD0.** A steady state with its grounding line (solid black contour) crossing the centre of the channel around $x = 450$ km was found by decreasing the spatially uniform rate factor to $A' = 1.5 \times 10^{-18}\,\text{Pa}^{-3}\,\text{a}^{-1}$. The resulting state is then close to the steady state without the damage model (i.e. with $\bar{\bar{D}} = 0$) and $A = 2.0 \times 10^{-17}\,\text{Pa}^{-3}\,\text{a}^{-1}$. The damage field grows from $\bar{\bar{D}} \approx 0$ through $\bar{\bar{D}} \sim 1/3$ around the grounding line to $\bar{\bar{D}} \sim 1/2$ in the lateral center of the shelf and $\bar{\bar{D}} \sim 1$ close to the domain boundaries. The lower panel plots a magnified portion of the upper panel, and shows damage increasing in the few kilometres upstream from the grounding line.

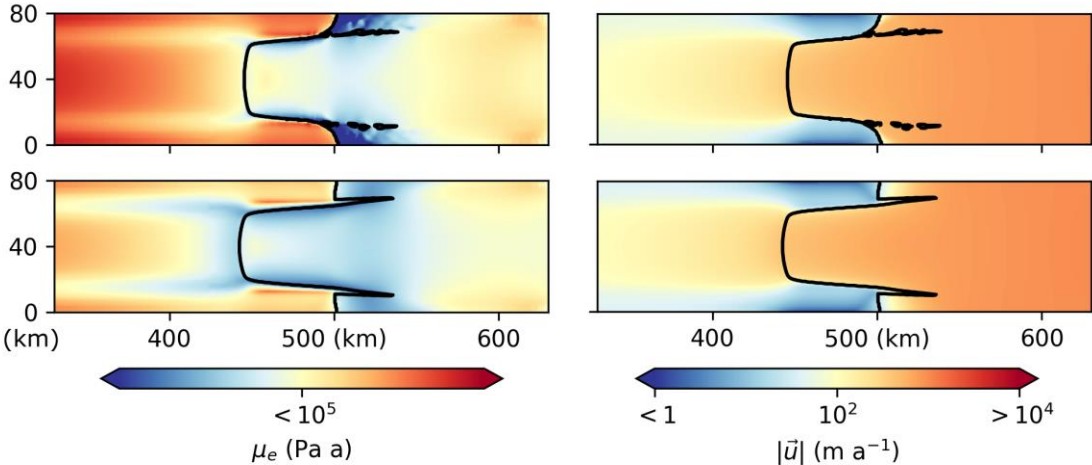

**Figure 4: Comparison of the final IceD0 (upper panel) and Ice0 (lower panel) patterns of vertically averaged effective viscosity ($\mu_e$) and speed ($|\vec{u}|$). The impact of damage on effective viscosity varies both laterally and between grounded and floating ice, but the shape and position of the grounding line (solid black contour) of IceD0 is similar to that of Ice0, as is the velocity field. These are approximately steady state patterns as the grounded area and volumes change very little after 1000 years (see also Figs 7 and 8).**

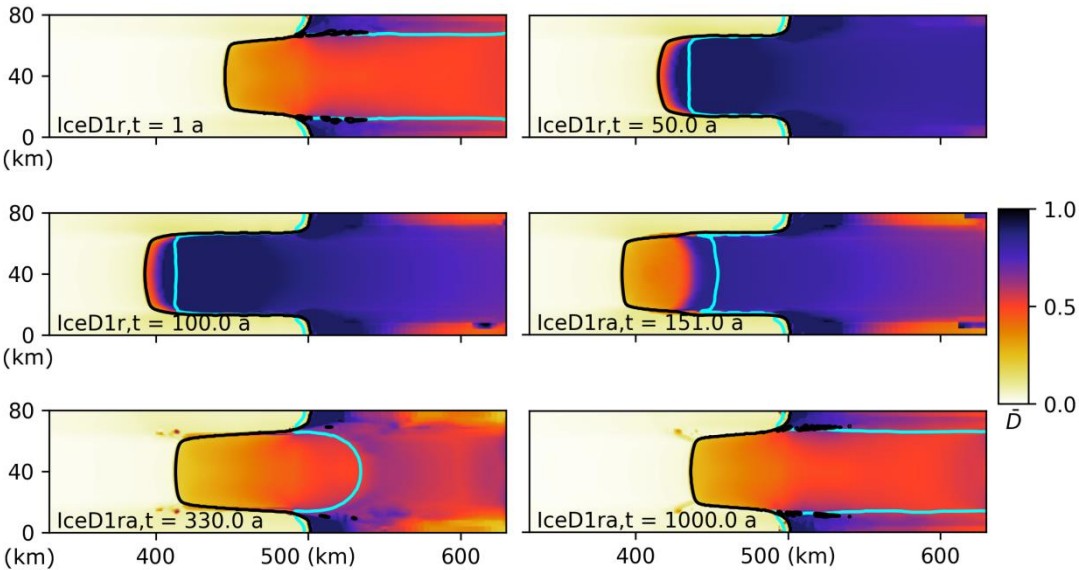

**Figure 5: Comparison of the retreat and recovery due to changes in ice shelf basal melt (IceD1rr and IceD1ra) in MISMIP+. IceD1r sees the grounding line (solid black contour) in the centre of the channel retreat to x = 390 km over 100 years in response to strong ablation across the whole ice shelf. Parts of the ice shelf are thin ( $h < 200$ m , outside cyan contour) so that a high value of $D$ implies only a small reduction in buttressing from those regions. IceD1ra follows directly from IceD1r, setting the melt rate to zero so that the ice shelf thickness and the grounding line re-advances, recovering its original shape after around 1000 years.**

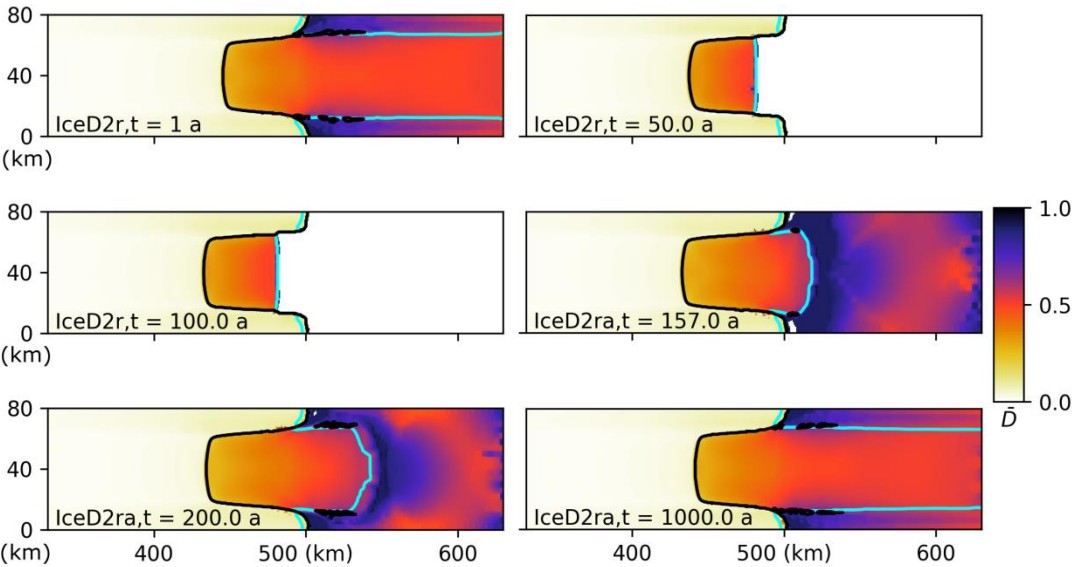

**Figure 6: Comparison of the retreat and recovery due to changes in calving front position (experiments IceD2rr and IceD2ra in MISMIP+).** Applying a melt rate of 100 m a$^{-1}$ where $x > 480$ km results in a remnant ice shelf whose damage field resembles the initial state in their common area, but grows somewhat toward the front (IceD2r). Loss of buttressing leads to grounding line (solid black contour) retreat. Once the melt rate is set to zero (IceD2ra) the calving front advances, carrying a region of elevated damage with it across the domain edge. Thin, damaged ice downstream from the $h = 200$ m contour (cyan) forms from direct accumulation on the sea surface.

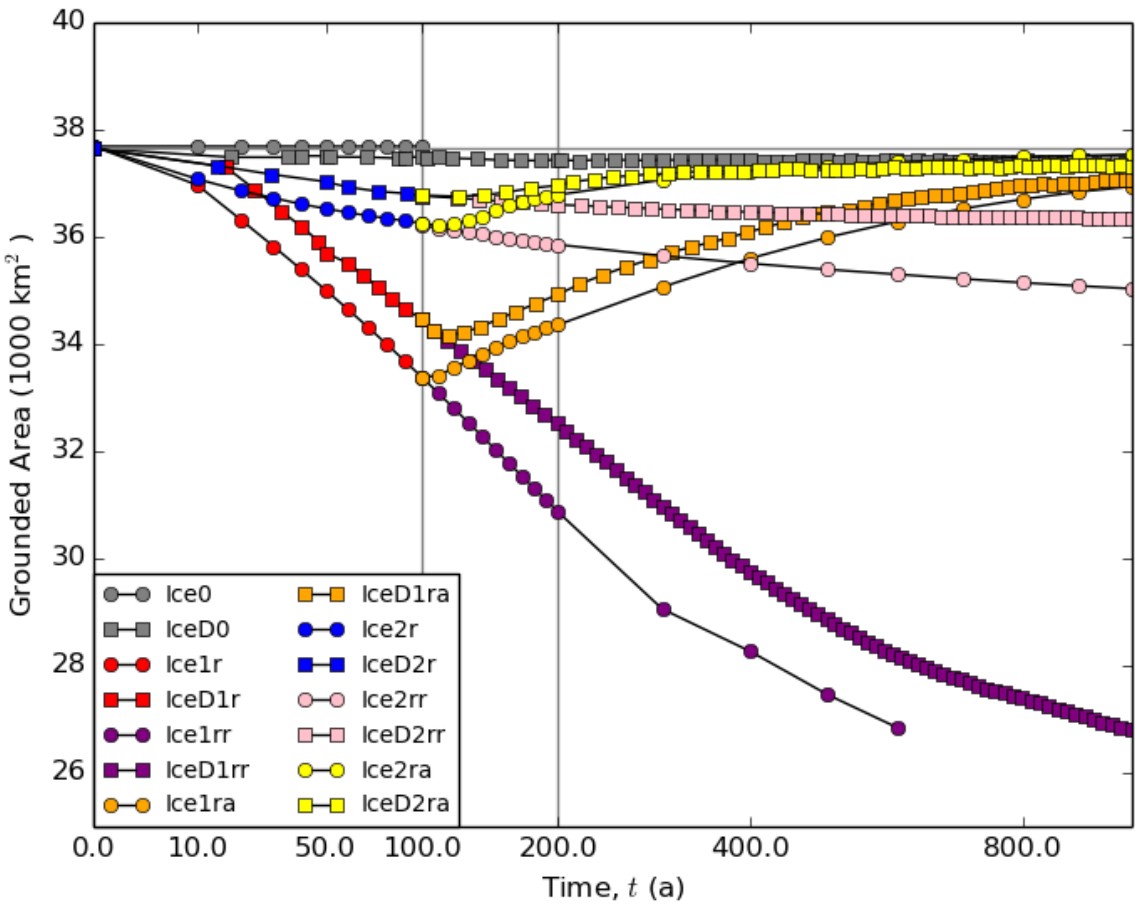

**Figure 7: Grounded area against time for all of the simulations. Lines with circles show the results based on BISICLES without the damage model and those with squares show the results based on BISICLES with the damage model. Grey curves show the results of the control run Ice0, IceD0; red curves show the results of Ice1r, IceD1r; orange curves show the results of Ice1ra, IceD1ra; purple curves show the results of Ice1rr, IceD1rr; blue curves show the results of Ice2r, IceD2r; yellow curves show the results of Ice2ra; IceD2ra; pink curves show the results of Ice2rr, IceD2rr. BISICLES_D produces slightly less retreat than BISICLES.**

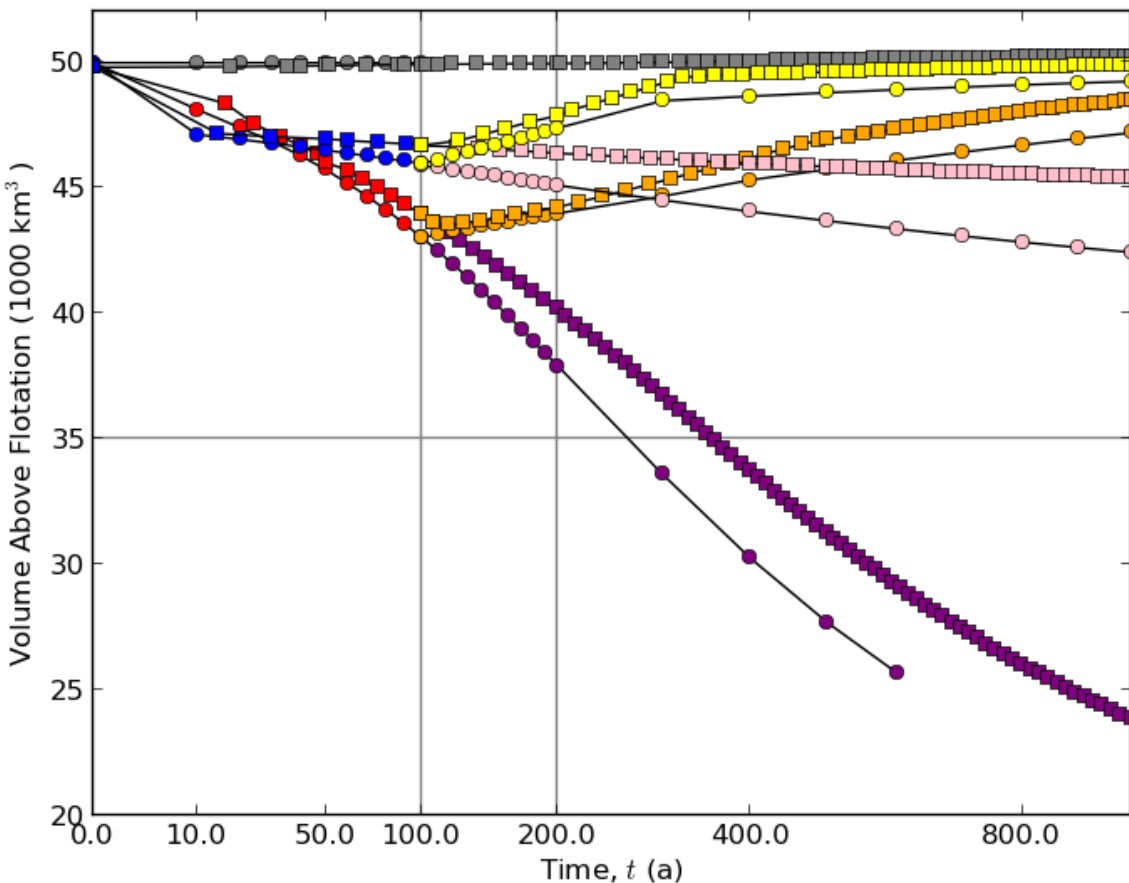

**Figure 8: As for Fig. 7 but for volume above flotation (VAF). VAF follows a similar trend as the grounded area (Fig. 7: it decreases rapidly when forced by melt rates, and increases more slowly to nearly the initial state when ocean forcing disappears. In both models, the ice sheet is more sensitive to basal melt near the grounding line than to extreme high basal melt representing calving at the front. However, BISICLES_D produces slightly less retreat under both ocean forcing scenarios than BISICLES.**

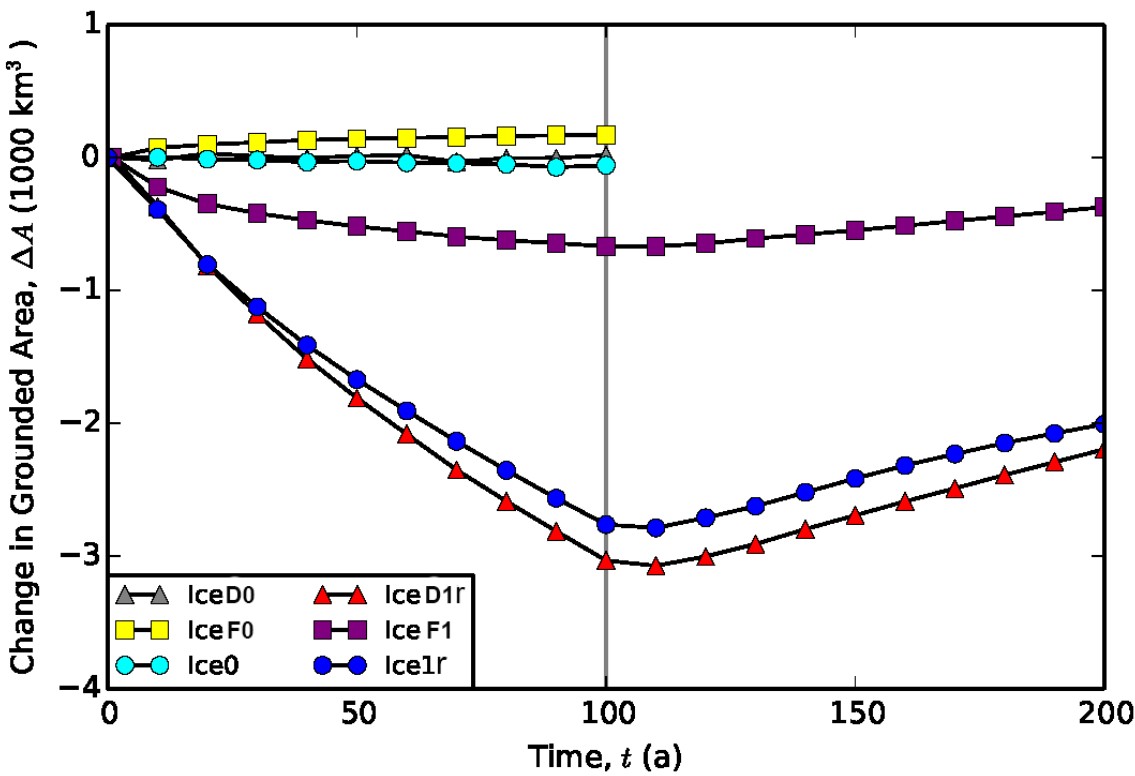

**Figure 9: Grounded area against time for the Ice0 (no forcing) and Ice1 (basal melting) experiments with no damage model (BISICLES) and the damage model (BISICLES_D) of Sect. 2.2 (IceD0, IceD1r) compared to a simulation where the damage field does not evolve with time but is taken from the initial state of IceD0 (IceF0, IceF1). In the no forcing experiments, the model with fixed damage (IceF0) produces a near-steady state close to those of Ice0 and IceD0. In the forcing experiments, the retreat of the grounding line in the fixed damage model (IceF1r) is much slower than in the model with evolving damage (IceD1r) and the model without damage and larger $A$ (Ice1r).**

| Experiment | Model | Viscous Parameter | Ocean forcing (1-100 year) | Ocean forcing (100-1000 year) |
|---|---|---|---|---|
| Ice0 | BISICLES | $A$ | 0 | 0 |
| Ice1ra | BISICLES | $A$ | Melting | 0 |
| Ice1rr | BISICLES | $A$ | Melting | Melting |
| Ice2ra | BISICLES | $A$ | "Calving" | 0 |
| Ice2rr | BISICLES | $A$ | "Calving" | "Calving" |
| IceD | BISICLES_D | $A$ | 0 | 0 |
| IceD0 | BISICLES_D | $A'$ | 0 | 0 |
| IceD1ra | BISICLES_D | $A'$ | Melting | 0 |
| IceD1rr | BISICLES_D | $A'$ | Melting | Melting |
| IceD2ra | BISICLES_D | $A'$ | "Calving" | 0 |
| IceD2rr | BISICLES_D | $A'$ | "Calving" | "Calving" |
| IceF0 | BISICLES-Fixeddamage | $A'$ | 0 | 0 |
| IceF1 | BISICLES-Fixeddamage | $A'$ | Melting | 0 |

Table 1: Summary of simulations carried out in the current study. The entries in the first column correspond to MISMIP+ experiment names (Asay-Davis et al., 2016).