# Peer review of "Ice shelf fracture parameterization in an ice sheet model"

_The Cryosphere, 2017_

## Referee Comment (RC1) · Anonymous Referee #1 · 3 May 2017

The authors have implemented a new representation of continuum damage mechanics into the ice dynamics model BISICLES, providing a way to study feedbacks between flow dynamics and damage-induced softening of the ice. To compare results from their modified model to standard results in the absence of damage, they follow the design of the MISMIP+ experiment. We believe that a number of points need to be addresses before publication. We also suggest additional experiments and an expansion of the discussions section, which could strengthen the manuscript.

In keeping with previously published papers, the authors adopt an advection scheme for damage. However, to the best of our knowledge, the authors suggest a new way to treat the source of damage in the advection equation, as detailed in section 2.2. The source is proportional to the local crevasse depth (surface + basal), where crevasse depths are calculated from a zero-stress criterion following (Nye 1957). We encourage

the authors to expand on the differences/parallels with existing research on continuum damage mechanics, to put their work into context, and to better motivate this approach. How and why is it "better" than previous work such as [Krug et al. 2014, Borstad et al. 2012, Pralong and Funk 2005]? For example, contrary to these studies, the authors do not implement a stress threshold for the formation of damage, and assume that non-zero damage is present in any tensile stress environment. Is this realistic, and how does it affect the results? Presumable the qualitative nature of the results remains the same, but it might become important at a later stage, when e.g. calving criteria are considered?

With regards to the results, the text provides an adequate description and explanation of the findings, although we would like to add a few comments/suggestions:

* The authors should specify how $d_w$ (crevasse water depth) in Eq. 8 is determined. Is it set to a constant value, and how is this value chosen?

* In Figure 8, a compelling argument is made that evolving damage could play an important role in simulating grounding line retreat/advance. However, the results are only discussed very briefly, which is disappointing. To strengthen their point, could the authors perform inversions for a spatially varying rate factor, using the surface velocity and geometry at different timesteps in the IceD0 and IceD1 experiments? It would be interesting to see how the rate factor changes over time, as one incorporates the effects of damage into its value. This could inform present-day model initialisation methods, as most models treat damage in the form of a spatially varying rate factor, which is kept constant in time.

* In order to increase the impact of this work, we suggest highlighting how the results have altered our understanding of damage, and indeed, whether it should be treated as a vital part of future ice flow modeling studies. Perhaps the authors could discuss in more detail the future directions of research (incl. possible calving laws?) they like to pursue, and whether this model can become a prognostic tool for calving?

[Figure]

\* Finally, we encourage native speakers on the author list to check the spelling and grammar of the manuscript.

Below is a list of more specific comments and questions about the text. (P refers to page number; L refers to line number on each page)

P2L5: "extremely sensitive to calving": I believe this statement could be misinterpreted as "more likely to calve". Therefore, please change the wording to "ice shelves in the Amundsen and Bellingshausen seas are thought to be more vulnerable in the event calving..." or similar.

P2L15: what do you mean by "magnify"? Do you mean that propagation and penetration of fractures causes calving?

P5L25: point out that A' is a constant, and not a spatially varying field

P6L1-3: The second question needs a better explanation. Perhaps write something along the lines of "If we adjust the rate factor such that the damaged model reaches a similar grounding line steady state compared to the undamaged model, how does the transient response between both setups differ, when subjected to an external forcing that leads to thinning of the ice shelf?

P6L12: reformulate this sentence as follows: "In order to start the MISMIP+ experiments from the required grounding line location at x=450m, we run a series of IceD simulations with different values of the rate factor A. For each value of A, a new steady state grounding line location is obtained, and we select the value A' for which the location is closest to the originally required grounding line at x=450m. We will refer to this steady state as IceD0."

P6L24: The reference to this table comes too late. Preferably refer the reader to this table before you start listing all the experiments, i.e. before line 5 on page 6.

P7L3: from hereon, the authors use capital letter D to refer to damage. Should this not be small letter d, in line with the definition in Eq. 10 as the vertical integral of the

damage?

P7L12: It is worth pointing out that a decrease in A leads to stiffer ice, making it intuitively easier to understand why this is the right thing to do.

P7L14: Reiterate that Figure 2 is for A' instead of A, and therefore the damage pattern looks different from Figure 1.

P7L14: Can you explain why the areas of high damage at the margins are not so well confined to narrow bands as in Figure 1?

P7L18: From the small figure it is unclear that the damage starts to grow a few kilometers upstream of the GL. Perhaps provide a zoomed-in version as an inset in Figure 2?

Figures 1-5: There is a lot of white space in all these figures that could be used to better display the details of your results. You should also consider choosing a different color to make the grounding line stand out better.

And a list of typos/suggestions where the text can be imporved. . .

P1L22: "former" -> replace by "previously"?

P1L24: ". . .Antarctica IN recent. . ."

P2L2: ". . .under THE present climate..."

P2L5: ". . .even A small amount. . ."

P2L6: "will trigger" is too strong, replace by "can trigger"

P2L13: "statistically continuum": what does this mean?

P2L17: ". . .based on THE calculation. . ."

P2L18: ". . .and THE calving rate. . ."

P2L22: reformulate sentence as follows: ". . .fields, and hence do not take into account

the stress history in the development. . ."

P2L24: "damage has AN effect on THE viscous behaviour. . ."

P2L29: "glacier'S"

P3L1: "state of art" -> replace by "state-of-the-art"

P3L6: ". . .the evolution of THE ice sheet, such as the speed and behavior of THE grounding line. . ."

P3L15: ". . .well in ice shelves. . ." -> ". . .well FOR ice shelves. . ."

P3L16: ". . .so given A bed elevation b and ice thickness h, THE surface elevation. . ."

P3L20: ". . .and THE two dimensional. . ."

P3L21-22: reformulate as ". . .is THE basal melt rate of the ice shelf. In equation (3), tr is the trace operator, E is the horizontal strain rate tensor. . ."

P4L2: remove "inland" as it is the same as "upstream"

P4L12: "proved" replace by "proven"

P5L12: ". . .represent THE effect of . . ."

P5L22: replace "sited" by "positioned"

P5L22: remove "towards the ocean"

P6L15: remove ". . .see the models respond..."

P6L24: remove "in real world"

P8L2: "extruds"??

P8L19: "floating" -> replace by "become afloat"

P8L19: ". . .and THE grounded area. . ."

P9L4: rewrite as "…The experiments Ice0 and IceD, which explicitly show the result of adding damage to the ice shelf, produced …"

P9L21: "as" -> replace by "at"

P9L25: rewrite as "…This does not mean that calving is unimportant for THE grounding line…"

P9L27: rewrite as "…the general case in reality, in particular for large ice shelves."

P10L4-5: remove excessive use of commas

P10L15: "In Biscles_D, THE viscosity…"

P10L16: "…we see THAT the retreat of THE grounding line…"

---

## Referee Comment (RC2) · J. Bassis (Referee) · 24 May 2017

**1 Overview**

This study incorporates a damage based parameterization of fracture in the BISICLES ice sheet model and uses this to assess the influence damage has on grounding line position. The BISICLES model is a sophisticated ice sheet model which includes mesh refinement. The goal of the present study is to examine the influence of damage on grounding line position using a MISMIP style setup. The authors introduce a damage formulation in which damage is determined based on the Nye crevasse depth formulation. This has the advantage that, unlike most damage evolution laws that are heuristically based on sparse laboratory or field measurements, damage evolution

has a physical component based on some elementary physics. Moreover, because crevasse depth models are popular methods of simulating the advance and retreat of outlet glaciers, the formulation has the potential to provide a unifying theme linking the behavior of outlet glaciers and ice shelves. The distinction between these two regimes is that damage in ice shelves is dominated by advection whereas damage in glaciers tends to grow rapidly near the calving front. In general, I think that this study is interesting and merits publication. However, I have a few major points that the authors should consider addressing in addition to several more minor nit-picky comments. The first sequence of questions relates to the physical formulation of the model where as the second relates to the overall structure of the manuscript and some difficulties I had working my way through it. Overall, however, the manuscript will be a valuable contribution to the field once these questions have been satisfactorily addressed.

**2 Major issues**

**2.1 Approach to damage**

The first comment that I have relates to the formulation of damage and advection of damage within the model. I like the general idea of the model and–this feels unseemly to point out in a review–proposed something similar several years ago (Bassis and Ma, 2015, Evolution of basal crevasses links ice shelf stability to ocean forcing, 10.1016/j.epsl.2014.11.003). There are, however, several key differences between the formulation proposed in that paper and in this one. In our model, we assumed that initial crevasse depths used to seed damage are determined by the Nye zero stress model, analogous to the model presented here. However, we used a perturbation analysis to examine how the crevasses evolve and in particular whether they deepen, widen or close. As we show in that paper, the evolution of the *ratio* of crevasse depth to ice thickness (a pseudo damage variable analogous to the one introduced in the present

study) is controlled by three factors. The first factor is simply kinematic. If crevasses are passive tracers in the flow field then they will deform with the flow field and their depth (or height) will decrease in exactly the same proportion as the ice thickness. A consequence is that the ratio of crevasse penetration to ice thickness remains *constant*. It is unclear to me how the kinematic distortion is accounted for here. From Equation (11) and (10) it looks like crevasse depths are inherited from upstream without accounting for the distortion associated with ice flow. This could be problematic. As we further show, the ambient stress field within the ice shelf will also result in crevasse growth or closure. In fact, crevasses are likely to widen, but penetrate a smaller portion of the ice thickness unless the tensile stress opening crevasses is larger than the stress for a freely spreading ice tongue. Again, this is based on a linear stability analysis and depends on the wavelength of the perturbation and is limited to the early stages of growth. Finally, we also show that the ratio of crevasse penetration to ice thickness will depend on the basal melt/refreezing regime of the ice shelf (for basal crevasses) or the surface mass balance (for surface crevasses). This again follows from kinematic considerations that depend on whether the melt/refreeze rates within crevasses is larger or smaller than the large-scale melt rate allowing the ocean to excavate crevasses or fill crevasses with marine ice. Again, it is unclear to me how the model proposed here accounts for these factors. To be clear we applied the formulation using observed ice shelf velocity and thickness fields as opposed to integrating it within an ice sheet model so our approach is not entirely transferable.

I'm also somewhat confused by the model used. This might be because symbols are introduced without definitions making it harder to follow the logic. For example, I have not been able to find a definition of $d_{tr}$. Similarly, I'm not sure I understand the right hand side of Equation 11. This seems to account for surface/basal mass balance, but it is introduced without explanatory text to help the reader understand the physics and assumptions.

There is also another subtle issue with the damage model proposed. In Bassis and

[Figure]

Ma (2015) we examined how individual crevasses would evolve using a perturbation analysis. The physical interpretation of damage here is more subtle. For example, suppose crevasses penetrate half of the ice thickness (or more generally X percent of the ice thickness) across a channel along the margin. Does that imply a channel cut into the ice shelf where the ice thickness is reduced by half? Does this also reduce the driving stress? Or are the crevasses assumed to be narrow so that they have little effect on the large-scale driving stress. In this case, the damage would then need to account for the fact that you have intact ice between crevasses, resulting in *lower* damage on a large-scale. Or perhaps crevasses are assumed to be filled with ice/melange? All of this is speculation and it would be helpful to have a cartoon or physical description of the process that readers can refer to.

**2.2   Organization**

I also struggled to understand the main hypothesizes tested. In the introduction we are told that the authors perform numerical experiments to address how including damage influences the evolution of the ice sheet and how the geometry of the damage field affects the dynamic response to ocean forcing. Later, at the beginning of Section 3 we are told that the goal is to address three question, including "If similar grounding line steady states can be realized with or without the damage model"; "If the 'hidden' damage inherent in the difference between A and A' is revealed in the response of the ice stream to thinning of the ice shelf" and; "If it is necessary to evolve the damage model in time or if one can get away with constructing a damage field at the start of a calculation and then merely hold it constant throughout the simulation." I don't object to any of the questions, but it would be helpful to have the main objectives of the study introduced together at the beginning. Perhaps the later three questions can be motivated as more specific versions of the initial questions? In fact, I'm not sure that all questions have been completely addressed–especially if the hidden damage is revealed by perturbation experiments. Perhaps I missed something. Nonetheless,

these five motivational questions would ideally also be mentioned in the abstract along with the resolution to the questions posed.

In a similar vein, one of the questions that authors seek to address is whether there is an equivalence between the rheology of damaged ice and ice with an adjusted rate factor A. The answer to this question seems obvious, especially when comparing Equations 5 and 6. We see that so long as we define $A' = A/(1 - D)^3$ there is an exact correspondence. That this question can be addressed by a simple mathematical definition makes me suspicious that the authors are examining a more subtle question, but if so it would help to provide more signposts for readers to help bring us along.

**3 Detailed comments**

The definition of 'damage' in Equations (6) and (7) doesn't follow naturally to me. In the standard approach to continuum damage mechanics one introduces a mapping from the actual stress $\sigma_{ij}$ to the effective stress $\tilde{\sigma}_{ij}$ of the form: $\tilde{\sigma}_{ij} = (1 - D)\sigma$. Note here that the mapping applies to the Cauchy stress tensor and not merely the deviatoric stress, as implied by Equation (6). It is true that one can define an effective viscosity of the form of Equation (6), but presumably one also must apply a mapping to the pressure term?

This leads me to my next question, typically the 'damage' is defined as a decrease in load bearing capacity associated with cross sectional area of micro-cracks within the ice. Hence, the damage takes on a value between zero and unity. Here damage is defined somewhat differently and damage is effectively unity everywhere there is a crevasse and zero elsewhere. Damage is thus binary instead of continuous. Upon depth integrating one obtains crevasse depth as the effective depth integrated damage variable. This new variable is no longer confined to the interval [0,1) and no longer behaves like a typical damage variable. However, one can define a new variable based on

the ratio of crevasse penetration depth (ds+db) to ice thickness H, which then maps the problem make to a more traditional effective damage variable that is again constrained to the interval [0,1]. This is what is done in Bassis and Ma (2015) and, as noted before, has several advantages in terms of the ability to account for kinematic distortion and passive advection.

Nonetheless, the authors use the variable "D" to denote what they call damage, which is nebulously defined, but appears to be three-dimensional, dimensionless and is binary taking on values of either 0 or 1. The authors then denote depth integrated damage using the lower-case "d" and this variable mimics crevasse penetration depth, which has units of length. Most of the model description focuses on depth integrated damage "d". However, starting in Section 4, the authors talk exclusively about damage with a capital D (see, e.g., page 7). Similarly, we see damage D in Figures 1,2, 4 and 5. This is acutely confusing because I thought that damage D was three-dimensional and binary and these figures all denote a single map view with a continuous variation in damage. I have a suspicion that the authors are really showing the ratio of crevasse penetration depth to ice thickness and these figures and much of the discussion is mislabeled, but I don't know for sure. Much of notation and discussion could be cleaned up to clear up reader confusion.

Page 2, Kachanov (1999) appears to be primarily based on metals and I am not aware of any observations presented therein that relate to ice. As such, I'm not sure that this is the best reference to support the hypotheses that micro-cracks are the ultimate source of crevasses. I think this is likely to be true, but one might thing about citing a more ice-centric study to support this.

Page 2, "calving rate increases with imbalance of forces". In the quasi-static approximation forces are always in balance. An imbalance of forces would result in acceleration and violate the Stokes flow hypothesis. Crucially, crevasses do not require an imbalance of forces to propagate.

Page 2, "The models discussed above have reasonably successfully reproduced the calving rate and fracture distribution on some individual glaciers." Is this really true? With the exception of Pralong and Funk, 2005, I don't think that damage mechanics have successfully predicted the calving rate on individual glaciers. This has always been in the hope of damage mechanics, but most of damage mechanics has so far been conceptually applied (e.g., Duddu and Waisman) or focused on the large-scale softening (e.g., Albrecht and Levermann, Borstad et al.,)

A complex bestiary of experiments. I had a very hard time keeping track of the bestiary of experiments and variants. The table is helpful and appreciated, but the authors can help readers a bit by reminding readers of key differences in the figure captions and providing a bit more explanation of what we are supposed to see in the figures. In general I prefer more descriptive figure captions that include a short figure title and then some more explanatory text to allow readers to quickly point readers to what the authors want to show.

There are quite a few typos in the manuscript (which I have not taken the pains to point out). The authors or the Cryosphere editorial staff need to give the manuscript a thorough proofreading.

---

## Author Comment (AC1) · 18 Jul 2017

**Response to referee 1: 'Ice shelf fracture parameterization in an ice sheet model' by Sainan Sun et al.**

**Received and published: 3 May 2017**

The authors have implemented a new representation of continuum damage mechanics into the ice dynamics model BISICLES, providing a way to study feedbacks between flow dynamics and damage-induced softening of the ice. To compare results from their modified model to standard results in the absence of damage, they follow the design of the MISMIP+ experiment. We believe that a number of points need to be addresses before publication. We also suggest additional experiments and an expansion of the discussions section, which could strengthen the manuscript.

**We would like to thank the referee for this detailed review.**

In keeping with previously published papers, the authors adopt an advection scheme for damage. However, to the best of our knowledge, the authors suggest a new way to treat the source of damage in the advection equation, as detailed in section 2.2. The source is proportional to the local crevasse depth (surface + basal), where crevasse depths are calculated from a zero-stress criterion following (Nye 1957). We encourage the authors to expand on the differences/parallels with existing research on continuum damage mechanics, to put their work into context, and to better motivate this approach. How and why is it better than previous work such as [Krug et al. 2014, Borstad et al. 2012, Pralong and Funk 2005]?

We think the main aim of the study is not so much to propose a new damage model, but to look at the impacts of implementing one versus neglecting it. To that extent, we chose a simple relationship between damage and tensile stress. If we consider only surface crevasses and neglect the threshold stress (more on that later), our model would be rather more similar to Krug et al 2014: they have a non-zero damage source when the Cauchy stress is positive, so as  $t \to \infty$  they will see  $D \to 1$  to the depth given by Nye zero-stress model, at least as far as vertically integrated models are a good approximation to Stokes models. Basal crevasess could be included in a model along the lines of Krug 2014 by adding the water pressure to the Cauchy stress, as in the Nye zero-stress model for basal crevases. Assuming that crevases open at least as quickly as the viscous deformation, we decided to compute the strain rate and damage simultaneously rather than by choosing some rate (as in Krug 2014).

We added a note on the timescale: "Specifying the damage-stress

relationship in this way assumes that damage evolves on a similar or faster timescale to the ice velocity field. Many authors specify instead a damage evolution rate, which, in Krug et al 2014, and given typical stresses in an ice shelf, amounts to a timescale of around 1 year." and a note on the similarity with Krug 2014 "Inasmuch as a shallow shelf model is a good approximation to the full Stokes model, our choice of the Nye zero stress model above is similar to the long-term behaviour of Krug et al 2014, at least for surface crevasses. In that model, damage grows where and only where the Cauchy stress is tensile, just as in the Nye model, giving the depth of surface crevasses. Basal crevassing could be included in such a model by adding the water pressure to the Cauchy stress, as in Keller and Hutter (2014)."

For example, contrary to these studies, the authors do not implement a stress threshold for the formation of damage, and assume that non-zero damage is present in any tensile stress environment. Is this realistic, and how does it affect the results? Presumable the qualitative nature of the results remains the same, but it might become important at a later stage, when e.g. calving criteria are considered?

Indeed, we do not impose a lower stress threshold for the formation of damge - this is in line with e.g the use of the same crevasse depth calculation in Nick 2010. Like them, we have assumed that the differences will be minor. At any rate, they will be limited to the upstream, slowing flowing parts of the ice where damage is low. We don't expect that such a threshold would make much difference to calving criteria, because we would only expect those criteria to be satisfied only in regions of large stress in either case. We added a note: "We also ignore any lower limit on the stress needed to open a crevasse, so that we will tend to produce small crevasses where there should be none. As we will see in the results, the major impact of the damage model is in the ice shelf and around the grounding line, where large tensile stresses readily exceed such limits".

With regards to the results, the text provides an adequate description and explanation of the findings, although we would like to add a few comments or suggestions:

The authors should specify how  $d_w$  (crevasse water depth) in Eq. 8 is determined. Is it set to a constant value, and how is this value chosen?

We added a note: A water depth  $d_w \sim h/2$  is in fact required in Nick et al (2010) for any calving to take place at all, and could clearly have a substantial impact on our calculations too, but for this paper we consider only dry surface crevasses, with  $d_w = 0$ .

In Figure 8, a compelling argument is made that evolving damage could play an important role in simulating grounding line retreat/advance. However, the results are only discussed very briefly, which is disappointing. To strengthen their point, could the authors perform inversions for a spatially varying rate factor, using the surface velocity and geometry at different timesteps in the IceD0 and IceD1 experiments? It would be interesting to see how the rate factor changes over time, as one incorporates the effects of damage into its value. This could

inform present-day model initialization methods, as most models treat damage in the form of a spatially varying rate factor, which is kept constant in time.

We did originally consider a set of synthetic inversions following the IceD1 experiment, but we would expect to simply recover the field D (or an approximation to it, with the difference being down to issues with the inverson method). Hence the experiment of figure 8 (now 9): in this case we do actually hold the damage constant in time as though it had been computed in an inversion at t = 0. We then see a much lower rate of retreat than either the full damage model (IceD1), or the original model with uniformly softer ice (Ice1). We made a poor job of describing and discussing this result in the original, and have re-written the relevant parts of both the result and the discussion sections, and added a short paragraph to the conclusion. As the reviwer suggests, this does have serious implications for present-day model initialization methods.

In order to increase the impact of this work, we suggest highlighting how the results have altered our understanding of damage, and indeed, whether it should be treated as a vital part of future ice flow modelling studies. Perhaps the authors could discuss in more detail the future directions of research (incl. possible calving laws?) they like to pursue, and whether this model can become a prognostic tool for calving?

We think the paragraph added to the conclusion summarises at least the immediate impact - modellers may need to reconsider their initilaization methods.

P2L5: "extremely sensitive to calving": I believe this statement could be misinterpreted as "more likely to calve". Therefore, please change the wording to "ice shelves in the Amundsen and Bellingshausen seas are thought to be more vulnerable in the event calving..." or similar. **Done.**

P2L15: what do you mean by "magnify"? Do you mean that propagation and penetration of fractures causes calving?

Yes we mean that. We change the word to 'trigger'.

P5L25: point out that A' is a constant, and not a spatially varying field **Done.**

P6L1-3: The second question needs a better explanation. Perhaps write something along the lines of "If we adjust the rate factor such that the damaged model reaches a similar grounding line steady state compared to the undamaged model, how does the transient response between both setups differ, when subjected to an external forcing that leads to thinning of the ice shelf? **Done.**

P6L12: reformulate this sentence as follows: "In order to start the MISMIP+ experiments from the required grounding line location at x=450m, we run a series of IceD simulations with different values of the rate factor A. For each value of A, a new steady state grounding line location is obtained, and we select the

value A' for which the location is closest to the originally required grounding line at x=450m. We will refer to this steady state as IceD0." **Done.**

P6L24: The reference to this table comes too late. Preferably refer the reader to this table before you start listing all the experiments, i.e. before line 5 on page 6.

**Done.**

P7L3: from here on, the authors use capital letter D to refer to damage. Should this not be small letter d, in line with the definition in Eq. 10 as the vertical integral of the damage?

We use small letter d to represent the vertical integral of the damage, but often it makes sense to refer to the vertical average, which we now denote  $\overline{D}(=d/h)$  to avoid the confusion caused by our earlier use of D.

P7L12: It is worth pointing out that a decrease in A leads to stiffer ice, making it intuitively easier to understand why this is the right thing to do. **Done.**

P7L14: Reiterate that Figure 2 is for A' instead of A, and therefore the damage pattern looks different from Figure 1. **Done.**

P7L14: Can you explain why the areas of high damage at the margins are not so well confined to narrow bands as in Figure 1?

The text should be "damage at the end of the IceD0 experiment". We modified now.

P7L18: From the small figure it is unclear that the damage starts to grow a few kilometers upstream of the GL. Perhaps provide a zoomed-in version as an inset in Figure 2?

Figure 2 (now 3) has been modified to include a zoomed in region around the GL as suggested, instead of showing the speed and effective viscosity, which appear in figure 3 (now 4).

Figures 1-5: There is a lot of white space in all these figures that could be used to better display the details of your results. You should also consider choosing a different color to make the grounding line stand out better.

We have remade all the figures with less white space, and in some cases fewer panels (which can then be larger). We changed the colormap for the damage field to one that is light for  $\overline{D} \approx 0$ , progressing through red and blu to black for  $\overline{D} \approx 1$ . This allows us to use a black line for the grounding line, and we then use cyan (which stands out well against red, dark blue, and black, even when printed in black and white) for the 200 m thickness contour.

And a list of typos/suggestions where the text can be imporved... P1L22: "former" ->replace by "previously"? **Done.**  P1L24: "...Antarctica IN recent..." Done. P2L2: "...under THE present climate..." Done. P2L5: "...even A small amount..." Done. P2L6: "will trigger" is too strong, replace by "can trigger" Done. P2L13: "statistically continuum": what does this mean? We delete the word 'statistically'. P2L17: "...based on THE calculation..." Done. P2L18: "...and THE calving rate..." Done. P2L22: reformulate sentence as follows: "...fields, and hence do not take into account he stress history in the development..." Done. P2L24: "damage has AN effect on THE viscous behaviour..." Done. P2L29: "glacier'S" Done. P3L1: "state of art" -> replace by "state-of-the-art" Done. P3L6: "...the evolution of THE ice sheet, such as the speed and behavior of THE grounding line..." Done. P3L15: "...well in ice shelves..." -> "...well FOR ice shelves..." Done. P3L16: "...so given A bed elevation b and ice thickness h, THE surface elevation..." Done. P3L20: "...and THE two dimensional..." Done. P3L21-22: reformulate as "...is THE basal melt rate of the ice shelf. In equation (3), tr is the trace operator, E is the horizontal strain rate tensor..." Done. P4L2: remove "inland" as it is the same as "upstream" Done. P4L12: "proved" replace by "proven" Done. P5L12: "...represent THE effect of..." Done. P5L22: replace "sited" by "positioned" Done. P5L22: remove "towards the ocean" Done. P6L15: remove "...see the models respond..." Done. P6L24: remove "in real world" Done.

P8L2: "extruds"?? P8L19: "floating" -> replace by "become afloat" Done. P8L19: "...and THE grounded area..." Done. P9L4: rewrite as "...The experiments Ice0 and IceD, which explicitly show the result of adding damage to the ice shelf, produced ..." Done. P9L21: "as" - > replace by "at" Done. P9L25: rewrite as "...This does not mean that calving is unimportant for THE grounding line..." Done. P9L27: rewrite as "...the general case in reality, in particular for large ice shelves." Done. P10L4-5: remove excessive use of commas Done. P10L15: "In BISICLES-D, THE viscosity..." Done. P10L16: "...we see THAT the retreat of THE grounding line..."

Done.

---

## Author Comment (AC2) · 18 Jul 2017

**Response to referee2 (J. Bassis): 'Ice shelf fracture parameterization in an ice sheet model' by Sainan Sun et al.**

**1 J. Bassis (Referee)**

Received and published: 24 May 2017

**1.1 Overview**

This study incorporates a damage based parameterization of fracture in the BISICLES ice sheet model and uses this to assess the influence damage has on grounding line position. The BISICLES model is a sophisticated ice sheet model which includes mesh refinement. The goal of the present study is to examine the influence of damage on grounding line position using a MISMIP style setup. The authors introduce a damage formulation in which damage is determined based on the Nye crevase depth formulation. This has the advantage that, unlike most damage evolution laws that are heuristically based on sparse laboratory or field measurements, damage evolution has a physical component based on some elementary physics. Moreover, because crevasse depth models are popular methods of simulating the advance and retreat of outlet glaciers, the formulation has the potential to provide a unifying theme linking the behavior of outlet glaciers and ice shelves. The distinction between these two regimes is that damage in ice shelves is dominated by advection whereas damage in glaciers tends to grow rapidly near the calving front. In general, I think that this study is interesting and merits publication. However, I have a few major points that the authors should consider addressing in addition to several more minor nit-picky comments. The first sequence of questions relates to the physical formulation of the model where as the second relates to the overall structure of the manuscript and some difficulties I had working my way through it. Overall, however, the manuscript will be a valuable contribution to the field once these questions have been satisfactorily addressed.

I would like to thank Dr. Bassis for this thorough review and I will try to give a response.

**1.2 Major issues:**

**1.2.1 Approach to damage:**

The first comment that I have relates to the formulation of damage and advection of damage within the model. I like the general idea of the model and this feels unseemly to point out in a review proposed something similar several years ago (Bassis and Ma, 2015 Evolution of basal crevasses links ice shelf stability to ocean forcing, 10.1016/j.epsl.2014.11.003). There are, however, several key differences between the formulation proposed in that paper and in this one. The work of Bassis and Ma, (2015) showed the instability effect of crevasses by strain rate weakening and the evolution of initially narrow crevasses. The penetration of crevasses depends on the the stress field and influenced by the basal melting or freezing in the crevasses. The work is definitely related to our study and we have cited it, noting in particular that some of its physics is missing from our model.

In our model, we assumed that initial crevasse depths used to seed damage are determined by the Nye zero stress model, analogous to the model presented here. However, we used a perturbation analysis to examine how the crevasses evolve and in particular whether they deepen, widen or close. As we show in that paper, the evolution of the \*ratio\* of crevasse depth to ice thickness (a pseudo damage variable analogous to the one introduced in the present study) is controlled by three factors. The first factor is simply kinematic. If crevasses are passive tracers in the flow field then they will deform with the flow field and their depth (or height) will decrease in exactly the same proportion as the ice thickness. A consequence is that the ratio of crevasse penetration to ice thickness remains \*constant\*. It is unclear to me how the kinematic distortion is accounted for here. From Equation (11) and (10) it looks like crevasse depths are inherited from upstream without accounting for the distortion associated with ice flow. This could be problematic.

Eq. (11)

$$\nabla \cdot (\mathbf{u}d_{tr}) = (\nabla \cdot \mathbf{u})d_{tr} + (\nabla d_{tr}) \cdot \mathbf{u} \tag{1}$$

does include this purely kinematic factor (the second term above) because  $(\nabla d_{tr}) \cdot \mathbf{u}$  is generally non-zero (with u being only the horizontal velocity). We added a note to the manuscript "The vertically averaged damage can be reduced through meteoric or marine ice accumulation, where ice thickens without an increase in  $d_{tr}$ . It can also be reduced or increased through the stretching and compression represented by  $(\nabla \cdot \mathbf{u})d_{tr}$ , in such way that the ratio  $d_{tr}/h$  remains constant"

As we further show, the ambient stress field within the ice shelf will also result in crevasse growth or closure. In fact, crevasses are likely to widen, but penetrate a smaller portion of the ice thickness unless the tensile stress opening crevasses is larger than the stress for a freely spreading ice tongue. Again, this is based on a linear stability analysis and depends on the wavelength of the perturbation and is limited to the early stages of growth.

The crevasses in our model don't have a width, and it seems that our vertically integrated model does not include this partcular effect. We have pointed that out in the manuscript.

Finally, we also show that the ratio of crevasse penetration to ice thickness will depend on the basal melt/refreezing regime of the ice shelf (for basal crevasses) or the surface mass balance (for surface crevasses). This again follows from

kinematic considerations that depend on whether the melt/refreeze rates within crevasses is larger or smaller than the large-scale melt rate allowing the ocean to excavate crevasses or fill crevasses with marine ice. Again, it is unclear to me how the model proposed here accounts for these factors. To be clear we applied the formulation using observed ice shelf velocity and thickness fields as opposed to integrating it within an ice sheet model so our approach is not entirely transferable.

The right side of eq. (11) defines the effect of surface mass balance and melting. Snowing at the surface can heal the crevasse, as can melt at the base (since the crevassed layer is eroded before the layer above). The melt rates are the same with in and out of the crevasses. We would need to do some further development to improve on this.

I'm also somewhat confused by the model used. This might be because symbols are introduced without definitions making it harder to follow the logic. For example, I have not been able to find a definition of dtr. Similarly, I'm not sure I understand the right hand side of Equation 11. This seems to account for surface/basal mass balance, but it is introduced without explanatory text to help the reader understand the physics and assumptions.

We have now defined the variable  $d_{tr}$ , "One more modification is needed to reflect the transport of damage by ice flow. At any one time and place we would have two fields, the  $d_l(x, y, t)$  computed above, and a field of transported crevasse depths  $d_{tr}(x, y, t)$  which would have originated at (x', y', t' < t) and been carried downstream, stretched, compressed, and so on." We have expanded the description of eq 11 to note that e.g basal melt is assumed to erode the crevassed lower layer so that verically integrated damge is reduced.

There is also another subtle issue with the damage model proposed. In Bassis and Ma (2015) we examined how individual crevasses would evolve using a perturbation analysis. The physical interpretation of damage here is more subtle. For example, suppose crevasses penetrate half of the ice thickness (or more generally X percent of the ice thickness) across a channel along the margin. Does that imply a channel cut into the ice shelf where the ice thickness is reduced by half? Does this also reduce the driving stress? Or are the crevasses assumed to be narrow so that they have little effect on the large-scale driving stress. In this case, the damage would then need to account for the fact that you have intact ice between crevasses, resulting in \*lower\* damage on a large-scale. Or perhaps crevasses are assumed to be filled with ice/melange? All of this is speculation and it would be helpful to have a cartoon or physical description of the process that readers can refer to.

In effect we are assuming that crevasses are filled with soft ice.

We have added a diagram, (fig 1) and some more text: "Notice that damage affects only the deviatoric stress (as in Jouvet et al, 2011 and Krug et al, 2014) and does not affect the gravitational driving stress. We might expect such a modification if we had instead modified the full Cauchy stress (as in Pralong and Funk, 2005, Bassis and Ma, 2015, and Mobasha et al 2016), but have assumed that damage has no impact with respect to isotropic compression or vertical shear, so that the usual hydrostatic vertical stress balance, and the usual vertical integral of the resulting horizontal pressure gradient holds. This is analogous to assuming that the crevasses are filled with an inviscid material having the same density as ice."

**1.2.2 Organization**

I also struggled to understand the main hypothesizes tested. In the introduction we are told that the authors perform numerical experiments to address how including damage influences the evolution of the ice sheet and how the geometry of the damage field affects the dynamic response to ocean forcing. Later, at the beginning of Section 3 we are told that the goal is to address three question, including "If similar grounding line steady states can be realized with or without the damage model"; "If the 'hidden' damage inherent in the difference between A and A' is revealed in the response of the ice stream to thinning of the ice shelf" and; "If it is necessary to evolve the damage model in time or if one can get away with constructing a damage field at the start of a calculation and then merely hold it constant throughout the simulation." I don't object to any of the questions, but it would be helpful to have the main objectives of the study introduced together at the beginning. Perhaps the later three questions can be motivated as more specific versions of the initial questions? In fact, I'm not sure that all questions have been completely addressed-especially if the hidden damage is revealed by perturbation experiments. Perhaps I missed something. Nonetheless, these five motivational questions would ideally also be mentioned in the abstract along with the resolution to the questions posed.

Our aim is to constuct a model that is amenable to large scale calculations, and to decide whether its impact on the ice flow justifies the further devleopment of such a model, or whether even simpler prescriptions (e.g, a rule of thumb for reducing A in the ice shelf) might be just as good. We have modified the manuscript in several places, including the abstract, to make this more obviuous.

In a similar vein, one of the questions that authors seek to address is whether there is an equivalence between the rheology of damaged ice and ice with an adjusted rate factor A. The answer to this question seems obvious, especially when comparing Equations 5 and 6. We see that so long as we define  $A' = A(1-D)^3$ there is an exact correspondence. That this question can be addressed by a simple mathematical definition makes me suspicious that the authors are examining a more subtle question, but if so it would help to provide more signposts for readers to help bring us along.

We meant some simple rule of thumb e.g A' = A/8 everywhere, rather than  $A' = A(1-D)^3$ , which would require knowledge of D(x, y, z) (or in our case d(x, y)) We did not make a good job of explaining this and have made a number of modifications to the text. One of the outcomes is that we do seem able to emulate the damage model with a simple prescription, at least in terms of the ice flow.

**1.3** Detailed comments:**

The definition of 'damage' in Equations (6) and (7) doesn't follow naturally to me. In the standard approach to continuum damage mechanics one introduces

a mapping from the actual stress  $\sigma_{ij}$  to the effective stress  $\tilde{\sigma}_{ij}$  of the form:  $\tilde{\sigma}_{ij} = (1 - D)\sigma_{ij}$ . Note here that the mapping applies to the Cauchy stress tensor and not merely the deviatoric stress, as implied by Equation (6). It is true that one can define an effective viscosity of the form of Equation (6), but presumably one also must apply a mapping to the pressure term?

Here we have followed others (e.g Krug 2014, Jouvet 2011) in only modifying the part of the stress that maps onto strain-rate. It does seem that a modification to the pressure term would be necessary if we were considering the ice to be weaker under isotropic compression, but we have assumed that it is not (i.e., crevasses are either closed but not bonded, or, as the reviewer suggests earlier, filled with incompressible melange.

This leads me to my next question, typically the 'damage' is defined as a decrease in load bearing capacity associated with cross sectional area of microcracks within the ice. Hence, the damage takes on a value between zero and unity. Here damage is defined somewhat differently and damage is effectively unity everywhere there is a crevasse and zero elsewhere. Damage is thus binary instead of continuous. Upon depth integrating one obtains crevasse depth as the effective depth integrated damage variable. This new variable is no longer confined to the interval [0,1) and no longer behaves like a typical damage variable. However, one can define a new variable based on the ratio of crevasse penetration depth (ds+db) to ice thickness H, which then maps the problem make to a more traditional effective damage variable that is again constrained to the interval [0,1). This is what is done in Bassis and Ma (2015) and, as noted before, has several advantages in terms of the ability to account for kinematic distortion and passive advection.

Nonetheless, the authors use the variable "D" to denote what they call damage, which is nebulously defined, but appears to be three-dimensional, dimensionless and is binary taking on values of either 0 or 1. The authors then denote depth integrated damage using the lower-case "d" and this variable mimics crevasse penetration depth, which has units of length. Most of the model description focuses on depth integrated damage "d". However, starting in Section 4, the authors talk exclusively about damage with a capital D (see, e.g., page 7). Similarly, we see damage D in Figures 1,2, 4 and 5. This is acutely confusing because I thought that damage D was three-dimensional and binary and these figures all denote a single map view with a continuous variation in damage. I have a suspicion that the authors are really showing the ratio of crevasse penetration depth to ice thickness and these figures and much of the discussion is mislabeled, but I don't know for sure. Much of notation and discussion could be cleaned up to clear up reader confusion.

The reviwer is entirely correct here. We made this clear in the model description "We construct a vertically integrated damage model by treating the ice sheet as having upper and lower layers of ice entirely fractured by surface and basal crevasses respectively, and an undamaged central layer (Fig, 1). Therefore, the scalar damage variable, D(x, y, z) employed in vertically varying models (Pralong and Funk, 2005; Jouvet et al, 2011; Keller and Hutter, 2014; Krug et al 2015; Bassis and Ma, 2015; Mobasha et al, 2016) takes on either the value

0 (in the central layer) or 1 (in the upper and lower layers). The principal damage variable in our model is  $d(x,y) \in [0,h)$ , the vertical integral of D(x,y,z), and our closest analogue to the usual D is its vertical average,  $\overline{D}(x,y) \in [0,1)$  "

Page 2, Kachanov (1999) appears to be primarily based on metals and I am not aware of any observations presented therein that relate to ice. As such, I'm not sure that this is the best reference to support the hypotheses that micro-cracks are the ultimate source of crevasses. I think this is likely to be true, but one might thing about citing a more ice-centric study to support this.

We cite Rist et al., (1994) now, which examines the relationship between microcracking and ice strength. The sentence is modified to: "Macro-scale fractures are originate from micro-scale cracks, which appear when viscous strain is too high (Rist et al., 1994)."

Page 2, "calving rate increases with imbalance of forces". In the quasi-static approximation forces are always in balance. An imbalance of forces would result in acceleration and violate the Stokes flow hypothesis. Crucially, crevasses do not require an imbalance of forces to propagate.

We intended to say "imbalance of forces could trigger crevasses propagation", but that's not the only reason. We now modified the sentence to:

"The crevasse depth and propagation depends on the stress field (Nye, 1957) and there are processes that further erode the ice, such as force imbalance and basal melting (Benn et al., 2007)."

Page 2, "The models discussed above have reasonably successfully reproduced the calving rate and fracture distribution on some individual glaciers." Is this really true? With the exception of Pralong and Funk, 2005, I don't think that damage mechanics have successfully predicted the calving rate on individual glaciers. This has always been in the hope of damage mechanics, but most of damage mechanics has so far been conceptually applied (e.g., Duddu and Waisman) or focused on the large-scale softening (e.g., Albrecht and Levermann, Borstad et al.,)

The description is inappropriate here. We delete the sentence.

A complex bestiary of experiments. I had a very hard time keeping track of the bestiary of experiments and variants. The table is helpful and appreciated, but the authors can help readers a bit by reminding readers of key differences in the figure captions and providing a bit more explanation of what we are supposed to see in the figures. In general I prefer more descriptive figure captions that include a short figure title and then some more explanatory text to allow readers to quickly point readers to what the authors want to show.

Agreed. We have added to the figure captions.

---

## Author Response (AR2)

**Response to referee 1: 'Ice shelf fracture parameterization in an ice sheet model' by Sainan Sun et al.**

**The referee is happy with this version of our manuscript. We would like to thank the referee for the review.**

**Response to referee 2: 'Ice shelf fracture parameterization in an ice sheet model' by Sainan Sun et al.**

This manuscript presents an updated version of a prior manuscript that sought to include damage in the BISICLES model. The original manuscript was promising, but had some glitches and confusing portions. The authors, however, have done a commendable job of thoroughly addressing reviewer comments. With the exception of one point of minor confusion (which might be my own), I only have nitpicky and minor comments.

**I would like to thank Dr. Bassis for the review and we will try to give a response.**

The one semi-important point I have is with Equation 11. Here the role of mass balance on damage is still unclear to me. It looks like the authors are assuming that snow deposited to the surface will increase the ice thickness, but will not affect the basal crevasse penetration depth. Geometrically, this makes sense if we imagine a basal crevasse. Snow deposited onto the surface then increases the ice thickness, but leaves the crevasse depth constant. In this case, snow "heals" the crevasse-even if this is a purely geometric effect. This prescription fails, however, if there is a surface crevasse, but this manuscript is mostly concerned with basal crevasses so this is not a significant problem. In contrast, melt around basal crevasses appears to erode the ice around crevasses and also decreases damage. This would seem to assume that there is no melt within the crevasse and all melt instead occurs outside of the crevasse. (Not an unreasonable assumption.)

**This is indeed our intention -- though we had thought that the healing term $(-\max(a, 0))\frac{d_{tr}}{h}$ made sense for surface crevasses, too, e.g with the same depth of ice deposited inside and outside a surface crevasse. As the referee says, it is really basal crevasses that matter in any case.**

This seems to be contracted, however, when the authors state that melt is assumed to be the same inside and outside of the crevasse. If the melt rate is the same inside and outside the crevasse, then it would appear as though the mass balance term should increase damage and by proportional to total melt rate (surface and bottom) and take the form $m \times \frac{d_{tr}}{h}$. See, for example, Equation 26 in Bassis and Ma (2015, but note the annoying typo where the r is missing from the second term). The reason the term appears this way is that if the melt rate is the same inside and outside the crevasses, the melt rate makes no

difference to the crevasse depth. The melt does, however, decrease the ice thickness. Hence, any (positive) melt rate will increase damage. It is possible that I have simply misread the authors statement, but I encourage the authors to check to make sure this statement isn't a typo and to make sure the equation is consistent with their description.

**The referee is correct here, and our sentence was careless - correct for accumulation at the surface but not melting at the base. We changed it to `Note that we do not attempt to include any additional accretion or ablation physics particular to the inside of crevasses'**

Some really nitpicky comments:

Near page 2 line 10: "Macro-scale fractures originate from micro-scale cracks, which appear when viscous strain is too high" Is this really what Rist says? Normally we think of micro-cracks forming when *stress* is high. Of course, stress and viscous strain rates are related through the rheology. The technical problem with the statement above is that at first glance, this implies that for a given stress, micro cracks are more likely to form in warm ice than cold ice because the viscous strain rates are larger. This is not well supported by the literature and, in fact, we often see that crevasses form when the stress (after temperature has been accounted for) exceeds a few hundred kPa. More pernicious, large portions of ice sheets have very large viscous **strains** because small strain rates have accumulated over millennia. This does not necessarily lead to micro-fractures because of other processes (like recrystallization). All in all, this is a minor point. If the authors want to be particular about it, I would probably say something along the lines of "Macro-scale fractures originate from micro-scale cracks, which are more likely to form when stresses within the ice become large" or if you want to be more specific "Macro-scale fractures originate from micro-scale cracks, which are more likely to form when stresses within the ice exceed a few hundred kPa"

**We take the suggestion of the referee and modify the sentence to be "Macro-scale fractures originate from micro-scale cracks, which are more likely to form when stresses within the ice exceed a few hundred kPa."**

Near page 2 line 10: "microscopic scales fracturing is a discrete process which operates on time scales determined by the speed of sound in ice" This is a fine statement. The authors might be interested to know that new research is, however, pointing out that slow rupture at speeds that are significantly less than body wave speeds is also possible. This can happen through a variety of processes including void growth and ductile failure. It is, however, unclear that any of these mechanisms are at work in ice.

**The sentence is modified to be "On microscopic scales fracturing is a discrete process which operates on time scales determined by the speed of sound in ice, and the rupture speed could be influenced by the local variations of stress state and material properties (Ye et al., 2016)."**

Results: breaking the results section down into subsections (perhaps identified by the different experiments) would help readers follow the different model results.

**We split this into sections now.**

Page 11, near line 5. I think there should be a tilde on the D?
**Yes, corrected.**

Also, it is interesting that D~0.5 appears to be an appropriate rule of thumb. This is, of course, the prediction for surface+basal crevasse depth that we would infer based on the Nye zero stress model in an unconfined ice tongue.
**Right, but we don't want to mislead here, so we have changed `$\bar{\bar{D}} \sim 1/2$ at all times in the ice shelf' to `$\bar{\bar{D}} \sim 1/2$ at all times in the ice shelf, with lower values ($\bar{\bar{D}} \approx 1/3$) close to the grounding line and in confined regions of the shelf'**